# LIPL-1 and LIPL-2 are TCER-1-regulated lysosomal lipases with distinct roles in immunity and fertility

Laura Bahr[1], Francis R.G. Amrit[1], Paige Emily Silvia [2], Bella Wayhs [1],
Guled Ali Osman[1], Mayur Nimbadas Devare [1], Hannah Henry[1], Danny Bui [1],
Mirae Choe [1], Nikki Naim[1], Margaret Champion[1], Yuxuan Man[1], Carissa Perez Olsen[2],
Arjumand Ghazi [1,3]*

1 Department of Pediatrics, University of Pittsburgh School of Medicine, John G. Rangos Sr. Research Center, One Children's Hospital Drive, Pittsburgh, Pennsylvania, United States of America, 2 Department of Chemistry and Biochemistry, Worcester Polytechnic Institute, Worcester, Massachusetts, United States of America, 3 Department of Cell Biology and Physiology, University of Pittsburgh School of Medicine, Pittsburgh, Pennsylvania, United States of America

* ghazia@pitt.edu

## Abstract

Reproduction and immunity are energy intensive processes that often compete for resources, leading to trade-offs across species. Lipid metabolism integrates these processes, particularly during stressful conditions such as pathogenic infections, yet the underlying molecular mechanisms remain poorly understood. TCER-1, the *C. elegans* homolog of mammalian TCERG1, suppresses immunity and promotes fertility, especially upon maternal infection. Here, we show that TCER-1 coordinates this balance by regulating two conserved lysosomal lipases, *lipl-1* and *lipl-2*. Using transcriptomic, lipidomic, and molecular-genetic analyses, we demonstrate that both lipases mediate infection-induced lipid remodeling but with distinct outcomes: *lipl-1* promotes immunity, whereas, *lipl-2* does not. LIPL-1 catalyzes the accumulation of specific ceramide species, including Cer 17:1;O2/24:0, whose supplementation rescues the immunity phenotypes of *tcer-1;lipl-1* mutants and enhances post-infection survival of wild-type animals. Both lipases influence fertility with *lipl-2* playing a key role in maintaining embryonic-eggshell integrity during maternal infection and aging. Remarkably, expression of human lysosomal acid lipase (hLAL/LIPA), the ortholog of '*lipl*' genes, restores immunity defects triggered by *lipl-1* loss and enhances immune resilience but does not significantly ameliorate the fertility defects. Together, these findings reveal distinct roles for *lipl-1* and *lipl-2* in modulating lipid species that link immune defense, reproductive fitness and healthspan through a potentially conserved mechanism.

**Data availability statement:** All relevant data are in the manuscript and its supporting information files. Raw RNAseq datasets have been deposited at NCBI and the accession information provided in S1 Table.

**Funding:** This work was supported by grants from the National Institutes of Health (R01AG051659, 1R56AG066682, R01AI176326, R21AG083329 to AG) and a Children's Hospital of Pittsburgh Research Advisory Committee (RAC) graduate fellowship (to LB). The funders had no role in study design, data collection and analysis, decision to publish, or preparation of the manuscript.

**Competing interests:** The authors have declared that no competing interests exist.

## Authors' summary

Our study sheds light on how the body balances two demanding processes- reproduction and immunity- that often compete for energy and nutrients. Using the nematode genetic model organism, *C. elegans*, we demonstrate that a gene called *tcer-1,* that encodes a transcription and splicing regulatory factor, helps manage this balance by controlling two fat-degrading enzymes, LIPL-1 and LIPL-2. These lipase enzymes have distinct roles: LIPL-1 boosts immune defense, while LIPL-2 predominantly supports fertility, particularly by keeping embryos healthy during maternal infection and aging. We found that LIPL-1 helps produce specific Ceramide fat species that protect against infection, and externally supplementing animals with these fats improved their survival upon bacterial infection. Interestingly, the human version of these enzymes, called lysosomal acid lipase A (hLAL/LIPA) can replace worm LIPL-1 to restore immune strength, suggesting this mechanism is conserved across species, but not the fertility-support roles. Overall, our work provides new insights into lipid metabolism's role in connecting immune resilience, reproduction, and longevity and reveals how organisms allocate energy to maintain health across different life stages.

## Introduction

Reproduction and immunity are intimately interconnected across species, from insects to humans, and often exhibit antagonistic interactions. In many species, infections reduce fertility, whereas, increased reproductive effort suppresses immune fitness [1,2]. However, exceptions exist as mating in some species triggers beneficial immune remodeling and heightened immune response [3,4], and male-derived factors can confer a survival advantage on females during post-mating infections [5]. In women, pregnancy was once considered to be a state of generalized immunosuppression but is now recognized to involve finely tuned immune adaptations that are essential from implantation through parturition, yet may also impair defense against certain pathogens [6–9]. Importantly, the intense metabolic demands of immunity and reproduction necessitate that these processes are tightly coregulated with energy metabolism [10,11]. Indeed, immune remodeling during pregnancy is partially driven by lipid metabolism, whereas, dysregulated energy metabolism underlies many pregnancy complications, including gestational diabetes and preeclampsia, the most common pathologies of human pregnancy in the developed world [12–14]. Thus, lipid metabolism emerges as a crucial arbiter of reproductive fitness and immune health coordination.

The molecular mechanisms linking lipid remodeling, immunity, and reproduction are poorly understood. In mammals, complexities of pregnancy, low fecundity, and long gestations hinder mechanistic studies, whereas, they have been more tractable to investigations in invertebrates [1,15]. The nematode model, *Caenorhabditis elegans,* offers unique advantages because its short generation time and

high reproductive output (~300 eggs over 3–6 days) allow sensitive detection of even minor fertility shifts under immune stress and *vice versa* [15–18]. For instance, maternal bacterial infection rapidly mobilizes lipids to the germline to support progeny but results in diminished maternal immune resistance [19–21]. Moreover, despite a simple immune system, *C. elegans* retains conserved immune signaling pathways, and many established infection models exist [22–27].

Previously, we identified TCER-1, *C. elegans* homolog of a mammalian transcription elongation and splicing factor, TCERG-1 [28,29], as a longevity-enhancing protein, that promotes fertility while suppressing immune responses in reproductively active animals [16]. Pertinently, TCER-1 supports reproductive success during immune challenges, as *tcer-1* loss-of-function (*lof*) mutants exhibit reproductive defects, whereas, TCER-1 overexpression partially mitigates the fertility loss associated with maternal infection [16]. TCER-1's role in coordinating reproduction, immunity, and aging appears to be evolutionarily conserved as inactivation of its *Arabidopsis thaliana* homologue, PRP40, enhances tolerance towards pathogens and delays flowering [30]. Mammalian *TCERG1* is highly expressed in reproductive tissues, and similar to *C. elegans* TCER-1, declines with age in mouse and human oocytes [31,32]. We showed that TCER-1, along with pro-longevity transcription factors, DAF-16 and NHR-49*,* increases lifespan upon germline loss through concomitant enhancement of lipid anabolism and catabolism [33,34]. We have now discovered that TCER-1 impairs immunity and promotes fertility, in part through regulation of lysosomal lipases, *lipl-1* and *lipl-2*, two of eight members of the conserved *'lipl'* lipase family orthologous to the lysosomal acid lipase (LAL/LIPA) in humans [35,36]. The *'lipl'* genes have been linked to nutrient stress responses, with *lipl-1* and *lipl-3* reported to mobilize fats during starvation [37], *lipl-2* and *lipl-5* to mediate metabolic remodeling upon dietary restriction [38–40] and *lipl-4* to promote longevity by activating a lysosomal-nuclear retrograde signaling pathway [41–43]. Notably, *lipl-1*, *lipl-2*, *lipl-3*, and *lipl-5* have been reported to be induced by pathogens such as *Enterococcus faecalis, Staphylococcus aureus, Cryptococcus neoformans,* and *lipl-2* is differentially expressed during *Bacillus thuringiensis* infection [43–45]. Yet, their functional roles in innate immune response remain unknown.

In this study, we show that TCER-1 regulates *lipl-1* and *lipl-2* expression upon infection by the human opportunistic Gram-negative pathogen, *Pseudomonas aeruginosa* strain PA14 (PA14) [27,46]. These lipases perform distinct functions in influencing immunity, reproduction, and longevity, and exert discrete effects on the animal's lipid composition. *lipl-1* enhances pathogen resistance, whereas *lipl-2* does not and, in fact, appears to limit immunity and longevity. Both lipases contribute towards fertility success, but *lipl-2* is especially critical for maintaining embryonic-eggshell integrity during maternal infection and aging. Notably, LIPL-1 and LIPL-2 influence the levels of individual lipid species that have been linked to stress resistance, lifespan, healthspan and reproductive outcomes. LIPL-1 drives the elevation of several specific ceramide species in *tcer-1* mutants, including Cer 17:1;O2/24:0, whose supplementation rescues the immunity phenotypes of *tcer-1;lipl-1* mutants and enhances post-infection survival of wild-type animals. Lastly, we show that human hLAL/LIPA rescues *lipl-1 lof*-associated immune defects and enhances worm survival upon PA14 infection but does not compensate for fertility phenotypes. Altogether, these findings provide evidence for discrete roles of LIPL-1 and LIPL-2 in *C. elegans* fertility and immunity, and as central effectors of TCER-1 in coordinating immune, reproductive, and aging processes.

## Results

### TCER-1 broadly remodels lipid metabolism in response to pathogenic infection in *C. elegans*

To elucidate the transcriptomes dictated by TCER-1 under uninfected conditions and upon pathogenic infection, we performed RNA-Seq on wild-type (WT) animals and *tcer-1(tm1452)* mutants that were exposed to PA14 for 8h and those maintained on standard *Escherichia coli* strain OP50 (OP50) lawns (Fig 1A). Analysis of the sequencing data using the CLC Genomics Workbench (S1 Table; see Methods for details) revealed that pathogen exposure triggered differential expression of 1002 genes in WT animals (Tab A in S1 Table) of which 583 were upregulated (Tab Ai in S1 Table) and 419 were downregulated (Tab Aii in S1 Table). *tcer-1* mutants showed differential expression of 159 genes as compared to WT on the normal OP50 diet (Tab B in S1 Table) with 78 being upregulated (Tab Bi in S1 Table) and 81 being downregulated

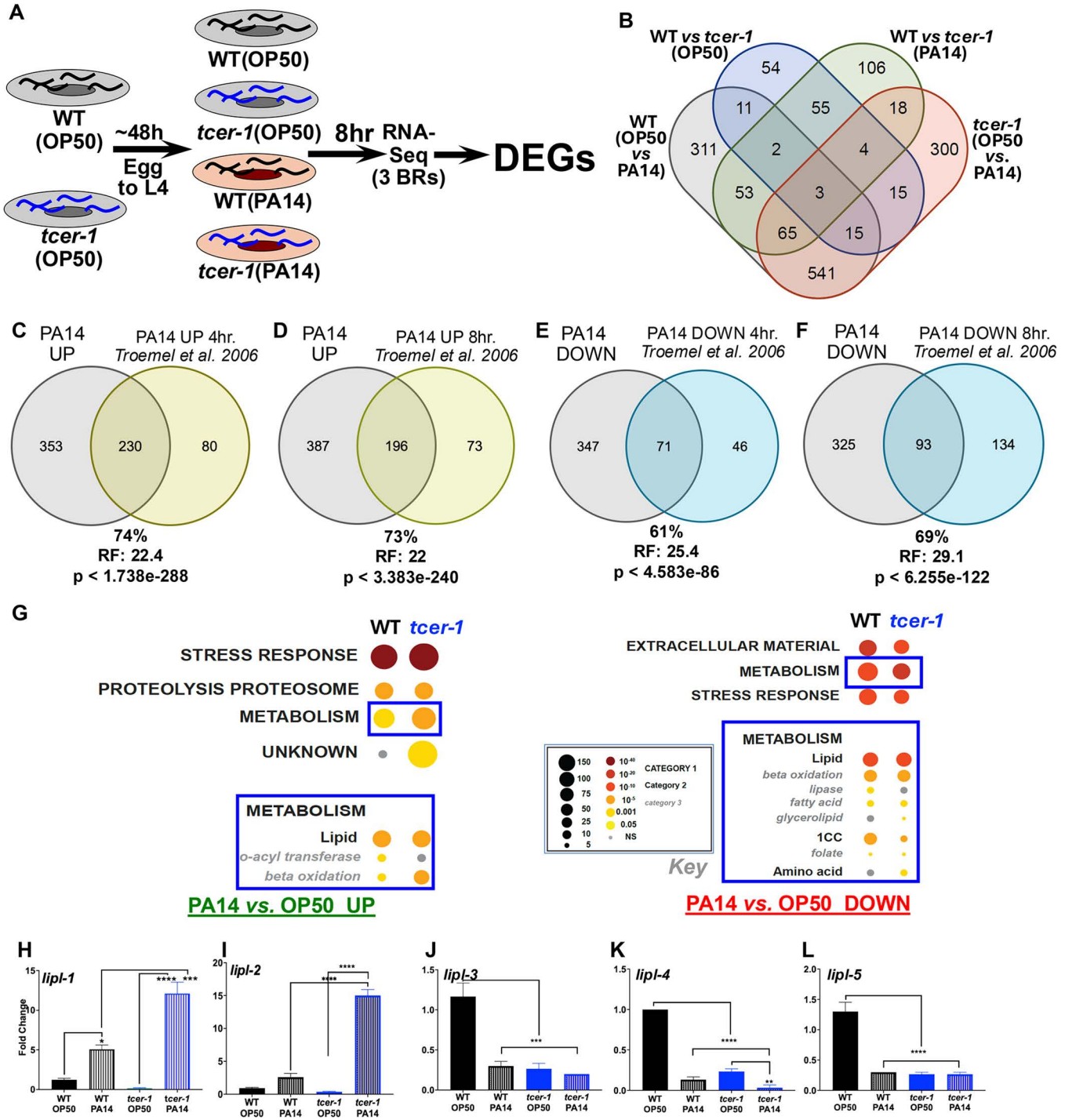

**Fig 1. TCER-1 regulates lipid metabolism upon PA14 infection. A) RNAseq experimental paradigm.** Age-matched wild type (WT) and *tcer-1* L4s raised on OP50 exposed to PA14 for 8 hours. **B) Overlaps of groups of differentially expressed genes (DEGs) identified. C-F)** Overlap of genes upregulated **(C, D)** and downregulated **(E, F)** upon PA14 exposure with previously-identified PA14-responsive genes by Troemel et al. 2006 [27]. RF: Representation Factor. Statistical significance of overlap between gene sets calculated using hypergeometric probability formula with normal approximation (see Methods). Comparisons with additional studies in S2 Fig and S2 Table. **G) Gene ontology (GO) term analysis of PA14- and TCER-1- driven DEGs using WormCat.** Metabolism (Category 1), particularly lipid metabolism, (Category 2) (blue boxes) identified as differentially impacted processes amongst genes upregulated (UP, left) and downregulated (DOWN, right) in infected WT and *tcer-1* mutants. Key shown in middle (black box). **H-L)** *lipl-1*

**and _lipl-2_ are transcriptionally upregulated in _tcer-1_ mutants on PA14 infection and _lipl-3, lipl-4_ and _lipl-5_ are downregulated.** mRNA levels of _lipl-1_ (**H**), _lipl-2_ (**I**) _lipl-3_ (**J**), _lipl-4_ (**K**) and _lipl-5_ (**L**) measured by qPCR in WT and _tcer-1_ mutant (blue) adults maintained on OP50 (solid bars) or exposed as L4s to PA14 for 8 h (hashed bars). Data from at least 3 independent trials/biological replicates. Asterisks represent statistical significance of differences observed in unpaired, two-tailed t-tests with P values * ≤ 0.05, *** p ≤ 0.001 and ****p ≤ 0.0001.

(Tab Bii in S1 Table). The expression of 306 genes was altered upon comparing WT animals and _tcer-1_ mutants after PA14 exposure (Tab C in S1 Table), of which 156 were upregulated (Tab Ci in S1 Table) and 150 were downregulated (Tab Cii in S1 Table). 962 genes were differentially expressed by _tcer-1_ mutants on OP50 _vs._ PA14 (Tab D in S1 Table) with 707 upregulated (Tab Di in S1 Table) and 255 downregulated (Tab Dii in S1 Table) (Fig 1B).

PA14-induced gene expression changes have been described in previous reports, and we found a substantial overlap between these datasets and our PA14-induced gene lists. For instance, 230 of 310 genes (74%) found to be upregulated 4 hours after PA14 infection by Troemel et al., were also upregulated in our study (Representation Factor (RF): 22.4, p<1.738e-288) as were 196 of 269 genes (73%) upregulated after 8 hours of infection (RF: 22.0, p<3.383e-240) (Fig 1C and 1D and S2 Table) [27]. 71 of 117 (61%; RF: 25.4, p<4.583e-86) and 93 of 134 (69%; RF: 29.1, p<6.255e-122) genes downregulated post-infection at 4 hours and 8 hours, respectively, in the same study were also downregulated on PA14 in our experiment (Fig 1E and 1F and S2 Table) [27]. Similarly significant overlaps were observed upon comparison with PA14-dependent genes identified by other studies (S1 Fig and S2 Table) [26,47]. These comparisons reinforced our confidence in the transcriptomes that we mapped, including the hundreds of newly- identified PA14 responsive genes, as well as the TCER-1 downstream targets.

Gene Ontology (GO) assessment and analyses with WormCat, a _C. elegans_ gene enrichment analysis tool [48], both revealed enrichment of innate immunity, stress response, and metabolic processes, particularly lipid metabolism, within the PA14- and TCER-1- dependent differentially expressed genes (DEGs) as previously reported (Figs 1G and S2 and Tabs A–D in S1 Table) [33,44]. Notably, five of the eight members of the _C. elegans_ 'lipl' family of lipase genes were affected by these interventions, forming two subsets with contrasting regulation. In WT animals, PA14 infection upregulated _lipl-1_ and _lipl-2_, whereas _lipl-3_, _lipl-4_, and _lipl-5_ were downregulated (Tabs Ai and Aii in S1 Table). These findings were confirmed by quantitative PCR (Fig 1H-1L). Importantly, _tcer-1_ mutants displayed a markedly greater induction of _lipl-1_ and _lipl-2_ upon infection compared to WT animals (Fig 1H and 1I). This trend was also reflected in the WormCat output. 'Metabolism' (Category 1) was enriched amongst DEGs of both WT and _tcer-1_ mutants' PA14 response, but the enrichment was stronger and more significant in _tcer-1_ mutants- mirroring the mutants' heightened induction of _lipl-1_ and _lipl-2_ (Figs 1G and S2). We focused on _lipl-1_ and _lipl-2_ because PA14 infection and TCER-1 appeared to exert opposite effects on their expression, and this contrasting regulation implied a potential role for these genes in the TCER-1- dependent tradeoff between immunity and fertility. In particular, the exaggerated upregulation of _lipl-1_ and _lipl-2_ in _tcer-1_ mutants suggested that in WT, infected adults these genes may be induced as part of the immune response but get curtailed by TCER-1, whereas, upon _tcer-1_ depletion, this repression is eliminated and immune response further heightened.

## TCER-1 differentially regulates _lipl-1_ and _lipl-2_ expression in adult tissues upon pathogen exposure

We next sought to identify the tissues and cells where _lipl-1_ and _lipl-2_ expression was being modulated. In a previous study, a 441 bp _lipl-1_ promoter-driven transgene was reported to be expressed in intestinal cells [37]. We created transgenic animals expressing mCherry driven by this promoter [_lipl-1p(441 bp):mCherry_]. Under uninfected conditions, _lipl-1_ expression was observed at low levels in WT adults with fluorescence localized to the intestine and, less frequently, in the head (Figs 2A and S3). Within the transgenic population, fluorescence intensity ranged from quite dim, and primarily visible in the posterior intestine, to very bright throughout the intestine and neurons. Most animals exhibited an intermediate level of fluorescence (Figs 2A-2F and S3). The fraction of worms showing high intestinal fluorescence was increased significantly upon _tcer-1_ RNAi as well as PA14 exposure (Fig 2A-2E), whereas, neuronal levels were not altered by either

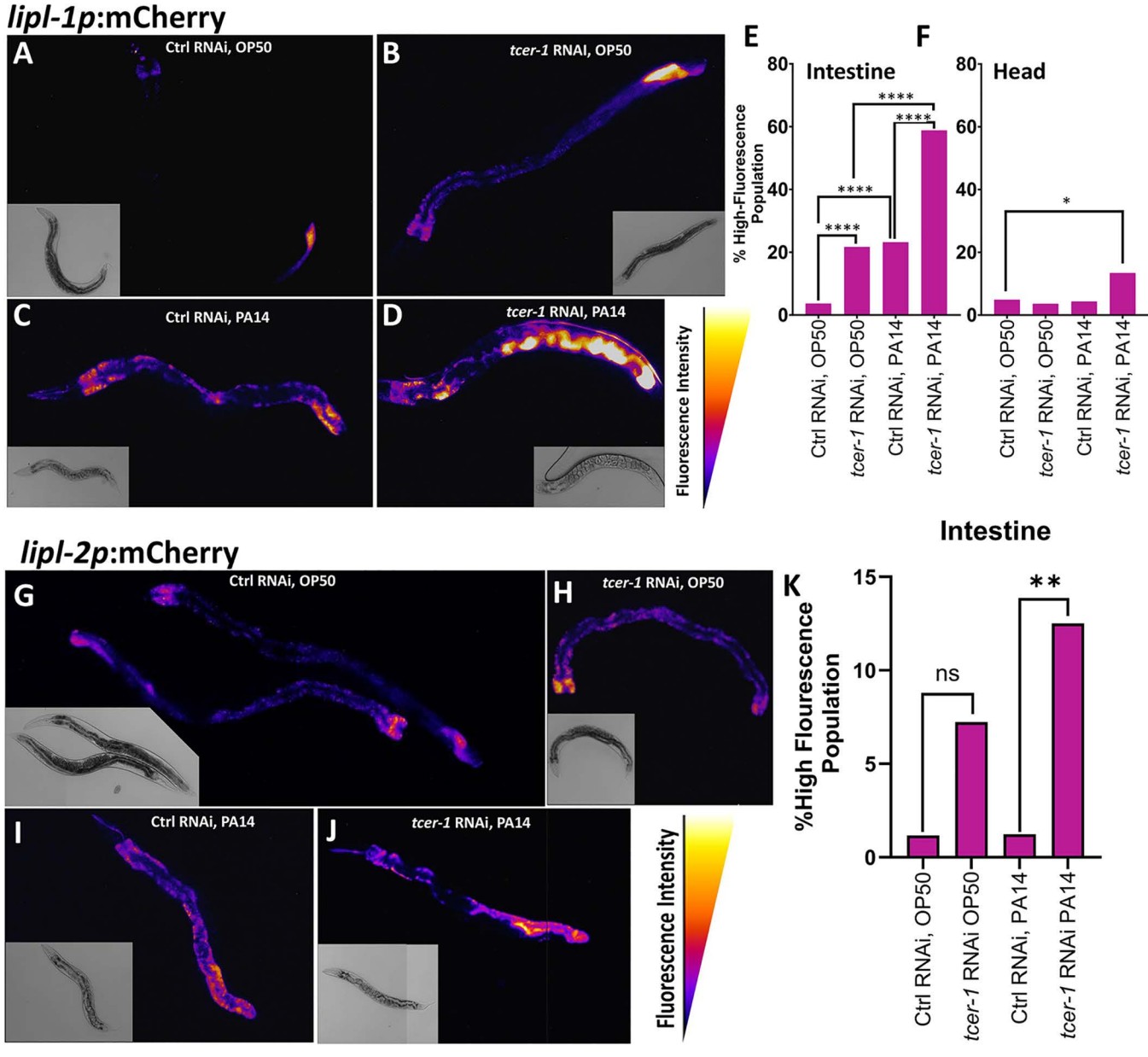

**Fig 2. *tcer-1* inactivation and pathogen exposure induce *lipl-1* and *lipl-2* transcriptional upregulation. A-F: *lipl-1* transcriptional changes visualized *in vivo* using *lipl-1p::*mCherry.** Animals raised on bacteria expressing control empty vector (**Ctrl, A, C**) or *tcer-1* dsRNA (**RNAi, B, D**) until preadult, L4 larval stage and transferred to plates seeded with *P. aeruginosa* PA14 (**PA14, C, D**) or *E. coli* OP50 (**OP50, A, B**) and incubated for 8h at 25 °C. Expression visible in intestine and head; in a WT adult population, varied from low levels to high or very high fluorescence intensities in intestine and head, respectively (categorization and quantification detailed in S3 Fig). Fraction of population with high or very high expression quantified in intestine (**E**) or head (**F**). **E:** Control (Ctrl) RNAi, OP50 (n = 81, 3.7%), *tcer-1* RNAi, OP50 (n = 83, 21.69%), Ctrl RNAi, PA14 (n = 69, 23.19%), *tcer-1* RNAi, PA14 (n = 68, 58.82%). **F:** Ctrl RNAi, OP50 (n = 81, 10.52%), *tcer-1* RNAi, OP50 (n = 83, 7.895%), Control RNAi, PA14 (n = 69, 7.895%), *tcer-1* RNAi, PA14 (n = 67, 23.68%). **G-K:** *lipl-2* **transcriptional changes visualized *in vivo* using *lipl-2p::*mCherry.** *lipl-2* expression levels observed using the transcriptional reporter *lipl-2p::mCherry*. Animals were raised as described above. Expression visible predominantly in the intestine and varied from low to high fluorescence intensities (detailed in S5 Fig). **K:** Fraction of populations showing high intestinal expression. Ctrl RNAi, OP50 (n = 86, 1.16%), *tcer-1* RNAi, OP50 (n = 83, 7.22%), Ctrl RNAi, PA14 (n = 81, 1.23%), *tcer-1* RNAi, PA14 (n = 72, 12.5%). A-D and G-J show representative images pseudocolored with ImageJ LUT Fire. Comparisons performed using two-tailed Fisher's exact test on contingency tables of data from 3 pooled biological replicates in both cases. *p ≤ 0.05, **p ≤ 0.01, ***p ≤ 0.001 and ****p ≤ 0.0001.

intervention (Fig 2F). Pathogen exposure following *tcer-1* RNAi tripled and doubled the fraction of animals exhibiting high expression in the intestines and neurons, respectively (Fig 2E and 2F) as predicted by the RNAseq data. We created an additional reporter strain driving mCherry under control of 1kb of *lipl-1* upstream region and found similar expression patterns (AGP340, see Methods) hence the 441 bp promoter reporter was used for subsequent studies. To examine how these transcriptional changes translated into protein levels and localization *in vivo*, we generated a transgenic strain expressing a fluorescent-tagged LIPL-1::RFP driven by the same 441 bp promoter. LIPL-1::RFP expression showed similar spatial distribution as the transcriptional reporter in intestinal cells, but at significantly lower levels. Expression in the head was punctate but highly inconsistent. Notably, expression was strongly localized to the coelomocytes (where the transcriptional reporter was not observed), suggesting that LIPL-1 protein may be secreted into the pseudocoelomic cavity (S4A-S4D Fig). While *tcer-1* RNAi and PA14 exposure both appeared to induce modest increases in RFP levels, parallel-ing the mRNA expression, we were unable to quantify these data due to the high variability and low expression levels of the fluorescent signal.

To examine *lipl-2* mRNA and protein localization *in vivo*, we similarly generated two mCherry-reporter strains driven by endogenous promoters (1kb and 1.5kb) and utilized a previously-published RFP-tagged strain (LIPL-2::RFP), respectively [39]. Both transcriptional reporters showed similar expression domains though the 1.5kb promoter-driven transgene showed more robust expression and was used to assess *lipl-2* regulation. *lipl-2* transcription was also most prominent in anterior and posterior intestinal cells with few animals showing head fluorescence (Figs 2G and S5). *tcer-1* RNAi produced a modest elevation of intestinal reporter expression, whereas, PA14 exposure had no impact. Upon PA14 exposure in animals subjected to *tcer-1* RNAi, *lipl-2* expression in the intestine was strikingly elevated as compared to animals on control RNAi (Figs 2G, 2J, 2K, S5A, and S5B). No changes in expression were observed in the head. Interestingly, and unlike *lipl-1*, *lipl-2* exhibited age-related expression dynamics with higher expression observed during larval stages followed by a marked decrease in young adulthood and a subsequent induction with age, by Day 5, in posterior intestinal cells (S5C Fig). LIPL-2::RFP sig-nal was very low in WT adults, localized primarily to the intestine and coelomocytes and with punctate, head fluorescence observed in a small number of animals; it was not visibly or quantifiably altered by PA14 or *tcer-1 lof* (S4E-S4H Fig).

## LIPL-1, but not LIPL-2, is essential for the enhanced pathogen resilience of *tcer-1* mutants

We sought to determine the physiological impact of *lipl-1* and *lipl-2* on the immune response against PA14. Survival experiments using publicly available, partial deletion mutants of the two genes yielded highly inconsistent results, so we used CRISPR-Cas9 to create complete deletion alleles of both genes (see Methods for details). We found that *lipl-1* was essential for the enhanced immune resistance of *tcer-1* mutants; in 4/5 trials, *tcer-1;lipl-1* mutants died significantly faster than *tcer-1* mutants and in 3 of these *tcer-1* mutants' enhanced resistance was completely abolished (Fig 3A and S3 Table). *lipl-1* single mutants' survival was not significantly different from that of WT animals (Fig 3A and S3 Table). Genes that enhance stress resistance, including immune stress, often increase longevity so we assessed the impact of *lipl-1* and *lipl-2* deletions on lifespan on the regular OP50 diet [49–54]. *lipl-1* deletion did not alter lifespan consistently, either alone or in a *tcer-1* mutant background (Fig 3B and S4 Table). *lipl-2* inactivation had unexpected, contrasting effects on immune resistance and longevity compared to *lipl-1*. The knockout did not suppress *tcer-1* mutants enhanced pathogen resistance and instead tended towards increasing it further (Fig 3C and S3 Table). *tcer-1;lipl-2* mutants showed lifespan extension on an OP50 diet compared to *tcer-1* mutants (Fig 3D and S4 Table). Together, these data suggest that *lipl-1* and *lipl-2* have distinct, context-dependent effects on immunity and longevity. *lipl-1* promotes immunity, especially upon *tcer-1* inactiva-tion, but does not affect lifespan, whereas, *lipl-2* suppresses immunity and longevity, especially upon *tcer-1 lof*.

## LIPL-1 and LIPL-2 are essential for embryonic eggshell integrity and reproductive fitness

Since TCER-1 promotes fertility, we examined the effects of *lipl-1* and *lipl-2* inactivation on reproductive fitness in *C. elegans*. The deletion of either *lipl-1* or *lipl-2* further aggravated the reduced progeny number (brood size) of *tcer-1*

## *lipl-1*

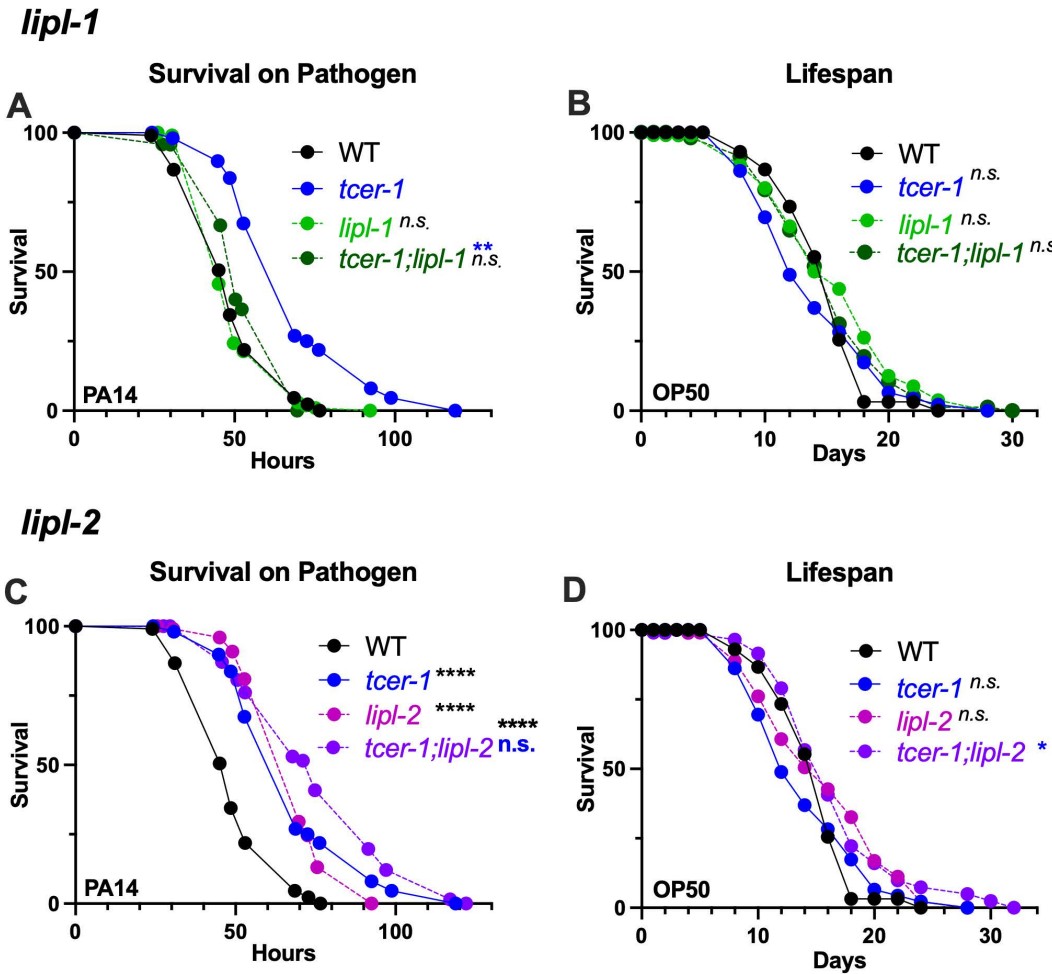

**Fig 3. *tcer-1* mutants' enhanced immunity is dependent upon *lipl-1* but not *lipl-2*. A, B: Impact of *lipl-1* deletion on survival upon pathogen infection and lifespan.** Survival of wild type (WT, black), *tcer-1* (blue), *tcer-1;lipl-1* (dark green) and *lipl-1* (light green) raised on OP50 till L4 stage and exposed to PA14 **(A)** or retained on OP50 **(B). A:** WT (m = 51.13 ± 1.32, n = 86/113), *tcer-1* (m = 70.61 ± 2.09, n = 87/112), *lipl-1* (m = 52.17 ± 1.1, n = 102/134), *tcer-1;lipl-1* (m = 56.04 ± 1.23, n = 85/120). **B:** WT (m = 14.96 ± 0.34, n = 94/121), *tcer-1* (m = 14.23 ± 0.48, n = 92/120), *lipl-1* (m = 15.79 ± 0.57, n = 80/106), *tcer-1;lipl-1* (m = 15.36 ± 0.51, n = 77/120). **C, D: Impact of *lipl-2* deletion on survival upon pathogen infection and lifespan.** Survival of wild type (WT, black), *tcer-1* (blue), *tcer-1;lipl-2* (grape) and *lipl-2* (magenta) raised on OP50 till L4 stage and exposed to PA14 **(C)** or retained on OP50 **(D). C:** WT (m = 51.13 ± 1.32, n = 86/113), *tcer-1* (m = 70.61 ± 2.09, n = 87/112), *lipl-2* (m = 70.64 ± 1.5, n = 61/123), *tcer-1; lipl-2* (m = 78.23 ± 2.66, n = 66/124). **D)** Lifespan on OP50. WT (m = 14.96 ± 0.34, n = 94/121), *tcer-1* (m = 14.23 ± 0.48, n = 92/120), *lipl-2* (m = 15.93 ± 0.56, n = 89/113), *tcer-1;lipl-2* (m = 16.72 ± 0.58, n = 81/103). Data from additional PA14 survival and lifespan trials in S3 and S4 Tables, respectively. Statistical significance was determined using log-rank Mantel Cox method and shown in each panel next to a given strain/condition. Asterisks are color coded to indicate the strain/condition being used for the comparison. p ≤ 0.05(*), *p* < 0.01 (**), < 0.001 (***), < 0.0001 (****).

mutants significantly (Fig 4A). Further, inactivation of either *lipl-1* or *lipl-2* induced sterility in *tcer-1* mutants (Fig 4B). *tcer-1;lipl-1* and *tcer-1;lipl-2* mutants also displayed a reduction in the number of eggs that hatched into healthy larvae (hatching) (Fig 4C). In *tcer-1;lipl-2 lipl-1* triple mutants, brood size was further diminished (Fig 4A), whereas, the impacts on sterility and hatching were not significant (Fig 4B and 4C). In the presence of a functional TCER-1, deletion of either *lipl-1* or *lipl-2* alone, or together, did not cause sterility or impair hatching (Fig 4B and 4C). However, a modest reduction in brood size was manifested in both single mutants which was not accentuated further in the *lipl-2 lipl-1* double knockout (Fig 4A).

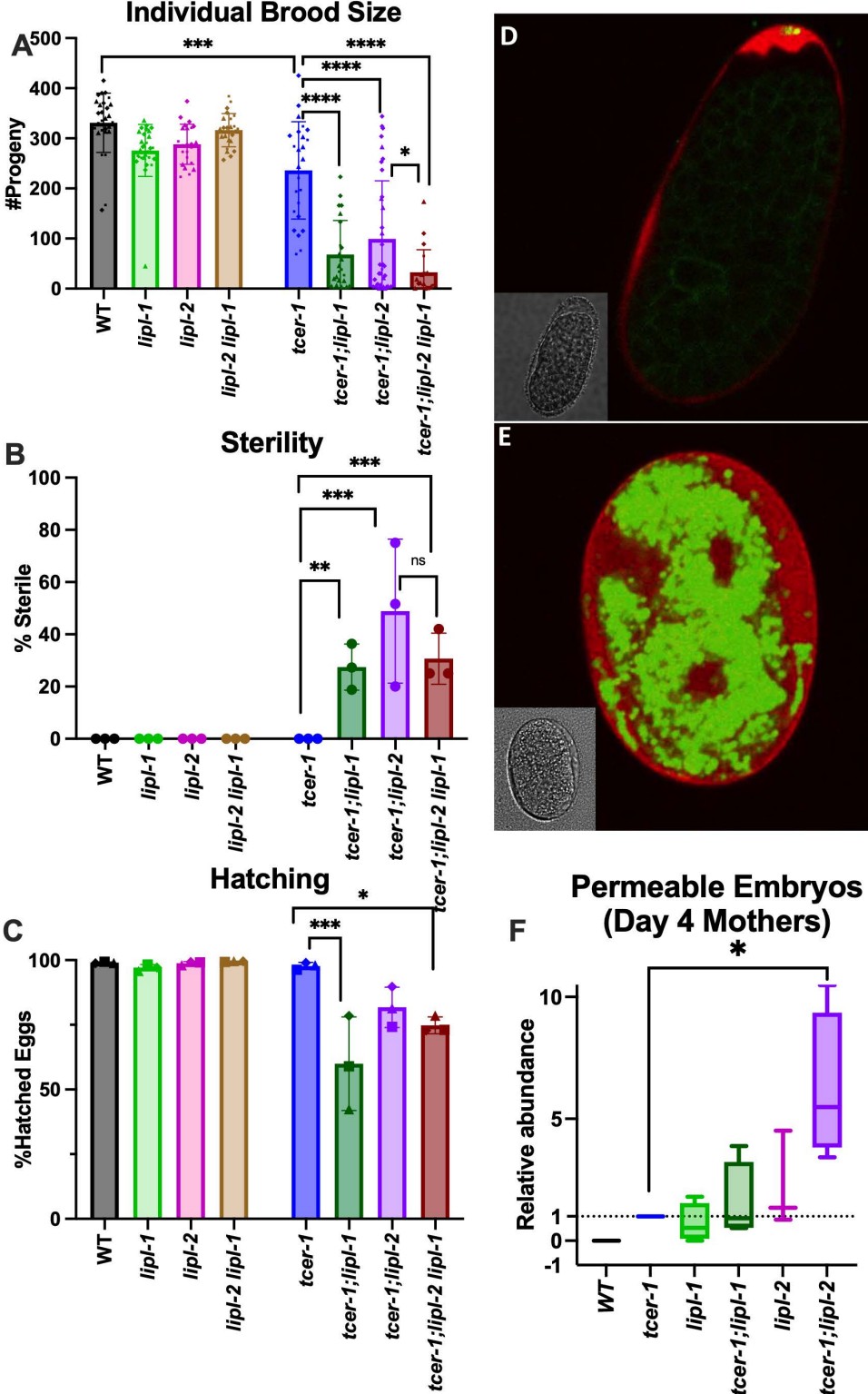

**Fig 4. *lipl-1* and *lipl-2 lof* result in decreased brood size, sterility, and embryonic inviability, particularly in combination with *tcer-1 lof*.** **A-C: Comparisons of maternal reproductive success** in wild type (WT, black), *lipl-1* (light green), *lipl-2* (pink), *lipl-2 lipl-1* (light brown), *tcer-1* (blue), *tcer-1;lipl-1* (dark green), *tcer-1;lipl-2* (grape) and *tcer-1;lipl-2 lipl-1* (dark brown) strains. **A: Brood size:** Total number of live progeny

counted per worm per strain. Each dot represents one worm from an aggregate of 3 independent trials. WT (n = 28, m = 330.9 ± 59.19), *lipl-1* (n = 29, m = 275.8 ± 51.97), *lipl-2* (n = 23, m = 288.2 ± 39.97), *lipl-2 lipl-1* (n = 24, m = 316.3 ± 33.23), *tcer-1* (n = 25, m = 236.2 ± 97.14) *tcer-1;lipl-1* (n = 23, m = 68.09 ± 68.12), *tcer-1;lipl-2* (n = 36, m = 99.31 ± 116.1), *tcer-1;lipl-2 lipl-1* (n = 20, m = 32.80 ± 44.87). **B: Sterility:** Percentage of animals which never produced a live progeny counted for each strain. WT (n = 28, m = 0), *lipl-1* (n = 29, m = 0), *lipl-2* (n = 23, m = 0), *lipl-2 lipl-1* (n = 24, m = 0), *tcer-1* (n = 25, m = 0), *tcer-1;lipl-1* (n = 23, m = 27.48 ± 8.82), *tcer-1;lipl-2* (n = 36, m = 48.87 ± 27.60), *tcer-1;lipl-2 lipl-1* (n = 20, m = 30.67 ± 9.81). **C: Hatching:** calculated from [#progeny/(#progeny+ unhatched eggs)]. WT (m = 99.01 ± 0.29), *lipl-1* (m = 97.09 ± 1.24), *lipl-2* (m = 98.81 ± 0.65), *lipl-2 lipl-1* (m = 99.47 ± 0.15), *tcer-1* (m = 97.75 ± 1.36), *tcer-1;lipl-1* (m = 59.95 ± 18.07), *tcer-1;lipl-2* (m = 81.75 ± 7.78), *tcer-1;lipl-2 lipl-1* (m = 74.79 ± 3.24). Data obtained from 3 independent trials in all cases. **D-F: Embryonic eggshell defects induced by *tcer-1, lipl-1* and *lipl-2* inactivation.** Representative images of embryos expressing *cpg-2p::mCherry::CPG-2* and *pie-1p::GFP::PH PLC1delta1*, labeling eggshell chondroitin proteoglycan layer and embryonic plasma membrane, respectively, incubated with lipid-labeling dye, BODIPY. **D:** BODIPY is excluded in healthy embryo with intact lipid-permeability barrier (LPB) and CPG-2 is sequestered away from the embryo, in a characteristic 'wavy' pattern in the perivitelline space of the eggshell. **E:** Embryo with defective LPB exhibits widespread BODIPY staining, and mCherry::CPG-2 which freely diffuses between the outer eggshell and the embryo. **F:** Quantification of BODIPY-permeable embryos. Average fold-change of the fraction of BODIPY stained embryos laid by Day 4 mothers, normalized to *tcer-1* mutants. *tcer-1* (n = 1964, m = 0), WT (n = 1930, m = 0), *lipl-1* (n = 2811, m = 0.7153 ± 0.7889), *tcer-1;lipl-2* (n = 1940, m = 6.222 ± 3.030), *tcer-1;lipl-1* (n = 2257, m = 1.563 ± 1.587), *lipl-2* (n = 2546, m = 2.244 ± 1.980). Data obtained from 4 independent trials in all cases except *lipl-2* for which 3 trials were conducted. In A-C, statistical significance was calculated using one-way ANOVA with Tukey's correction. In F, student's t-test was used. $p \leq 0.05$(*), $p < 0.01$ (**), $< 0.001$ (***), $< 0.0001$ (****).

Next, we examined the integrity of the embryonic eggshell, a protein and lipid rich structure critical for normal growth and development as it protects the embryo from mechanical and osmotic disruptions [55,56]. In particular, the innermost layer of the eggshell, a fat-rich lipid-permeability barrier (LPB) serves as a deterrent against unregulated influx of external materials [55]. Hence, WT embryos are impermeable to lipid-staining dyes such as BODIPY or FM-64; when LPB is disrupted, the eggshell becomes permeable to these molecules and fluorescence can be observed within the embryonic body (Fig 4D and 4E) [57,58]. Embryonic porosity towards lipid dyes serves as a valuable measure of LPB and eggshell integrity. In WT animals, we did not find any embryos that were permeable to BODIPY when laid by young, Day 1 mothers, or reproductively older Day 4 ones (Figs 4F and S6). However, embryos laid by Day 4 mothers carrying *tcer-1* or either *'lipl'* deletion produced a small but consistent fraction of eggs that accumulated BODIPY. The percent of porous eggs was not significantly increased in the *tcer-1;lipl-1* double mutants. In contrast, this population was enhanced by up to tenfold in *tcer-1;lipl-2* mutants (Fig 4F). Indeed, *lipl-2* and *tcer-1;lipl-2* mutants not only laid BODIPY-penetrable eggs on Day 4 of adulthood but also on Day 1 (S6 Fig). Altogether, these data suggested that both lipases promote fertility in WT animals and *tcer-1* mutants, with *lipl-2* exerting a greater impact, and their absence leads to defective eggshell formation that may contribute to reduced embryonic viability and fertility defects.

## TCER-1 and LIPL-2 Influence the relative distribution of neutral lipid populations

To decipher the molecular changes brought about by TCER-1 on the animals' lipid profile, and assess the contributions of LIPL-1 and LIPL-2, we performed high performance liquid chromatography coupled with tandem mass spectrometry (HPLC-MS/MS) on young Day 1 WT adults and age-matched *tcer-1*, *tcer-1;lipl-1* and *tcer-1;lipl-2* strains as well as *lipl-1* and *lipl-2* single mutants. HPLC-MS/MS allowed us to identify the impact of the genes on the abundance and relative distribution of major neutral lipid (NL) and phospholipid (PL) classes as well as individual lipid species within those classes. Within the global NL population, *tcer-1* mutants exhibited a significant shift in the relative abundance of the triacylglyceride (TAG) *vs.* diacylglyceride (DAG) levels. Compared to WT animals, their relative TAG content was increased by 15–20%, whereas, DAG levels were concomitantly decreased (Fig 5A and 5B). Interestingly, *lipl-2* single mutants showed a similar change in TAG:DAG distribution. *tcer-1;lipl-2* double mutants showed a small increase as compared to either genes' single mutant but this did not achieve statistical significance. In contrast, *lipl-1* inactivation had little impact in either WT or *tcer-1* mutant backgrounds (Fig 5A and 5B) suggesting that, amongst the two lipases, *lipl-2*, but not *lipl-1* shaped the TAG:DAG distribution.

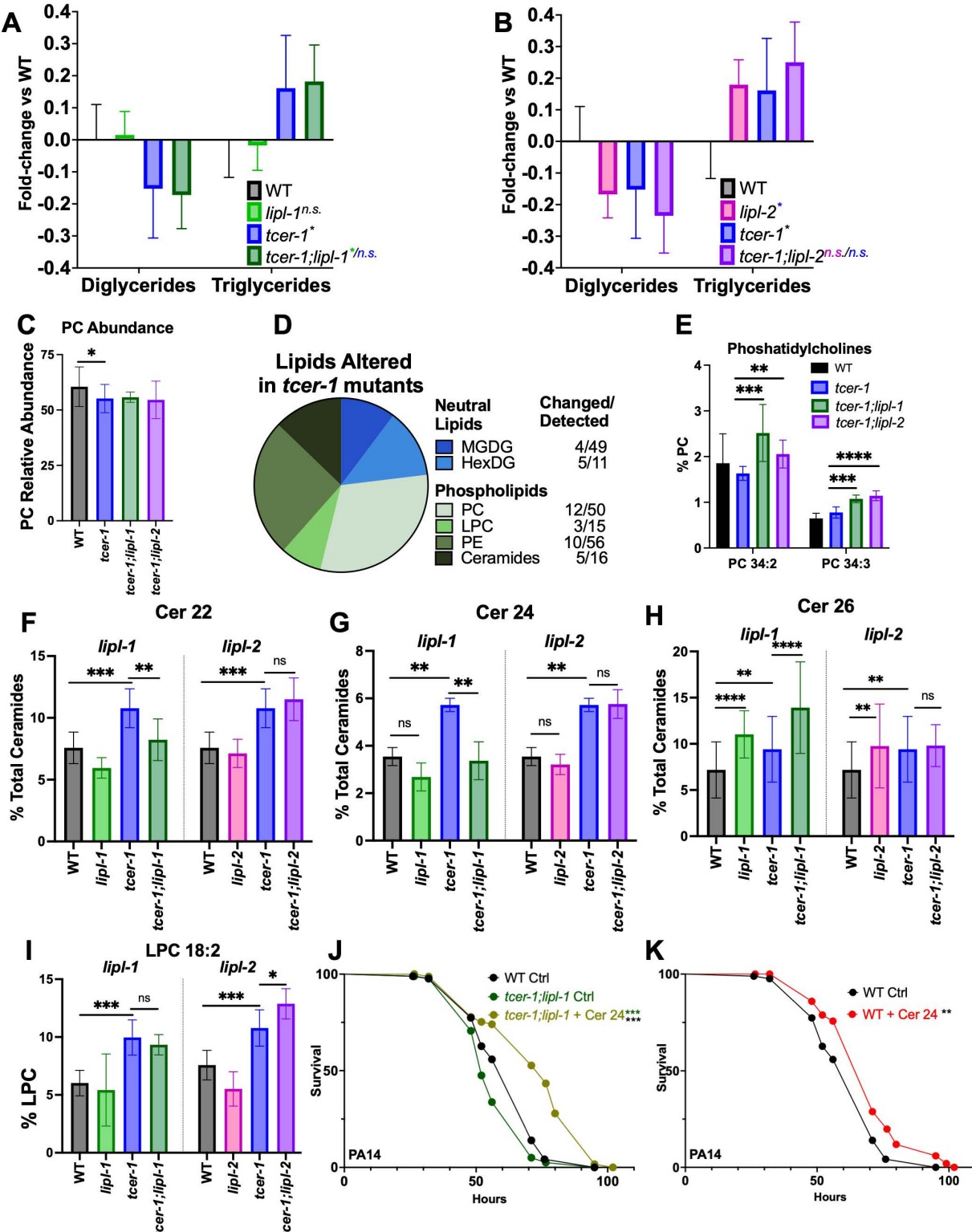

**Fig 5. *lipl-1* and *lipl-2* have both shared and distinct impacts on the lipidome of *tcer-1* mutants. A-C: Effects of *lipl-1* (A) and *lipl-2* (B) loss on broad neutral lipid (NL) and phospholipid (PL) populations** in wild type (WT, black) and *tcer-1* mutant (blue) animals. Bars indicate the fold-change of each data point normalized to the WT average. Statistical significance calculated using mixed-effects analysis with Tukey's correction. **C:** Relative

abundance of overall phosphatidylcholine (PC) levels. **D:** Categorization of NL (top, blue) and PL (bottom, green) species whose abundance is altered in *tcer-1* mutants. For each class, number of species identified and those changed in *tcer-1* mutants shown in the table. **E:** Relative abundance of PC 34:2 and PC 34:3 compared to all PCs in different strains. **F-I: Distinct impacts of *lipl-1* (F-H) and *lipl-2* (I) on lipidome of *tcer-1* mutants. F-H:** Relative abundance of Cer 17:1;O2/22:0 (Cer 22) **(F)**, Cer 17:1;O2/24:0 (Cer 24) **(G)** and Cer 17:1;O2/26:0 (Cer 26) **(H)** compared to all ceramides. **I:** Relative abundance of lysophosphatidyl choline (LPC) 18:2 compared to all LPCs. Data derived from HPLC-MS/MS analysis of Day 1 young adult hermaphrodites isolated in 5–6 independent biological trials. Statistical significance was calculated using two-way ANOVA with Tukey's correction. **J-K: Ceramide supplementation enhances survival upon PA14 infection.** Survival of wild type Ctrl (WT, black), *lipl-1;tcer-1* ctrl (green), *lipl-1;tcer-1* + Cer 24 (Olive), WT + Cer 24 (Red) raised on OP50 till Day1 and exposed to PA14. **J:** WT ctrl (m = 63.09 ± 1.53, n = 74/90), *lipl-1;tcer-1* ctrl (m = 58.1 ± 1.33, n = 82/90), *lipl-1;tcer-1* + Cer 24 (m = 73.74 ± 2.0, n = 71/90). **K:** WT ctrl (m = 63.09 ± 1.53, n = 74/90), WT + Cer 24 (m = 70.57 ± 1.78, n = 58/90). Data from additional PA14 survival trials in S7 Table. Statistical significance was determined using log-rank Mantel Cox method and shown in each panel next to a given strain/condition. Asterisks indicate the statistical significance and their color the strain being used for the comparison. p ≤ 0.05(*), *p* < 0.01 (**), < 0.001 (***), < 0.0001 (****), n.s: not significant.

### LIPL-1 and LIPL-2 impact the levels of phospholipid species implicated in stress resistance and healthspan

In contrast to NLs, the relative abundance of the major PL classes was not significantly altered in *tcer-1* mutants except for a small but significant reduction in the levels of phosphatidylcholines (PC) (Figs 5C and S7). However, *tcer-1* mutants manifested changes in the abundance of specific PL species' head groups. Of the 39 lipid species significantly altered in *tcer-1* mutants compared to WT animals, 30 were PLs and 9 were NLs. Amongst PLs, PCs were the most affected class as 12 of 50 identified species were altered in *tcer-1* mutants- 7 were decreased and 5 increased (Fig 5D and S5 Table). Besides PCs, phosphatidylethanolamines (PE) were the most disrupted (10 of 56 species identified) followed by lyso-phosphatidylcholines (LPCs) and ceramides (Cer) (Fig 5D and S5 Table). While neither *lipl-1* nor *lipl-2* deletion affected the total PC abundance in *tcer-1* mutants, single mutants of both genes showed a small but significant reduction in overall PC levels compared to WT (S7C and S7D Fig) and influenced the overall saturation of the associated fatty acid chains in the NL and PL populations (S8 and S9 Figs). *lipl-1* and *lipl-2* had shared impacts on the levels of two PCs, PC 34:2 and PC 34:3, both of which were significantly elevated in the *tcer-1;lipl-1* and *tcer-1;lipl-2* double mutants as compared to the *tcer-1* single mutants (Fig 5E and S5 Table). Notably, *lipl-1* and *lipl-2* also exerted distinct effects on *tcer-1* mutants' lipidome, most strikingly on the Cer and LPC compositions. *tcer-1* mutants showed increased levels of three Cer species, C17:1;O2/22:0 (Cer 22), Cer 17:1;O2/24:0 (Cer 24) and Cer 17:1;O2/26:0 (Cer 26) (Figs 5F-5H and S10 and S5 Table). The elevation of Cer 22 and Cer 24 was significantly reduced in *tcer-1;lipl-1*, whereas, Cer 26 levels were further elevated. *tcer-1;lipl-2* mutants showed levels of all three species like *tcer-1* (Figs 5F-5H and S10 and S5 Table). LIPL-2 had a similarly distinct impact on the abundance of LPC 18:2, which has been identified as a biomarker of healthspan and lifespan in human and *C. elegans*, respectively [59,60]. LPC 18:2 levels were significantly enhanced in *tcer-1* mutants (Fig 5I and S5 Table). While *lipl-2* single mutants showed a small reduction compared to WT, in *tcer-1;lipl-2* mutants it was further elevated compared to *tcer-1* alone. *lipl-1* inactivation had no impact in either genetic background (Fig 5I and S5 Table). The impact of *lipl-2* was specific to LPC 18:2 as two other LPCs, 17:1 and 19:1, that were depleted in *tcer-1* mutants compared to WT, were not impacted by *lipl-2* deletion (S5 Table). *lipl-1* and *lipl-2* single mutants also manifested overlapping as well as distinct lipidomic changes compared to WT (S6 Table). Altogether, these data showed that LIPL-1 and LIPL-2 had shared as well as discrete impacts on the lipidome, under normal conditions and upon *tcer-1* inactivation. Importantly, they revealed the influences of these lipases on specific lipid moieties found previously to be associated with longevity and healthspan outcomes.

### Cer 24 supplementation extends the post-infection survival of *tcer-1;lipl-1* mutants and WT adults

The effects on Cer 22 and Cer 24 paralleled the requirement of *lipl-1* for the enhanced PA14 resistance of *tcer-1* mutants, suggesting potential immunity-promoting functions for these species. Amongst the Cer species elevated in *tcer-1;lipl-1* mutants, we were able to commercially procure Cer 24 for supplementation studies. Exposing Day 1 *tcer-1;lipl-1* adults to 60µg Cer 24 for 24 h before, and during, PA14 infection enhanced their survival substantially as

compared to control animals exposed to solvent alone (Fig 5J and S7 Table) indicating a physiological relevance to the LIPL-1-mediated production of this species. We next asked if Cer 24 benefited WT animals and found their post-infection survival was also extended significantly upon Cer 24 supplementation (Fig 5K and S7 Table). To assess the roles of LIPL-2-derived species similarly elevated, we tested if LPC 18:2 improved WT survival on PA14 but found no impact upon its supplementation (S7 Table).

### Human lysosomal acid lipase (hLAL/LIPA) rescues immune deficits, but not the reproductive phenotypes induced by *lipl-1* and *lipl-2* mutations

*C. elegans* LIPLs are predicted to be orthologs of several human lipases, including hLAL/LIPA that is expressed in mammalian immune cells such as macrophages [61] and is implicated in macrophage dysfunction, inflammatory signaling and lysosomal storage diseases [35,62]. To assess a potential functional homology between *C. elegans* LIPL-1 and LAL, we used MosSCI technology to express hLAL/LIPA in *C. elegans* and asked if the human protein could substitute for LIPL-1 in its immune function. We found that, upon PA14 infection, expression of hLAL in *tcer-1;lipl-1* mutants completely rescued their survival to *tcer-1* level (Fig 6A and S8 Table). In addition, transgenic strains expressing hLAL in somatic cells of WT *C. elegans* also survived substantially longer upon PA14 exposure (Fig 6B and S8 Table) suggesting a conserved immunity-promoting function. We next asked if hLAL/LIPA could rescue the fertility phenotypes associated with either *lipl-1* or *lipl-2* inactivation in *tcer-1* mutants. These results were equivocal. hLAL/LIPA expression did not improve the brood-size defects of *tcer-1;lipl-1* (Fig 6C). *tcer-1;lipl-2* mutants expressing hLAL/LIPA showed a trend towards enhanced brood size compared to the double mutant alone, but this did not achieve statistical significance (Fig 6D). Similarly, the *tcer-1;lipl-2* mutants' embryonic permeability defects were marginally ameliorated but this was not statistically significant either (Fig 6E) indicating that hLAL/LIPA can substitute for the immunity-promoting *'lipl'* function but likely not the fertility roles.

## Discussion

In this study, we identified *lipl-1* and *lipl-2* as mechanistic effectors of TCER-1, a transcriptional and splicing regulator that suppresses immunity and promotes fertility in *C. elegans*. Through transcriptomic, lipidomic, and genetic analyses, we show that these conserved lysosomal lipases perform distinct, context-specific roles in shaping immunity, reproduction, lipid profile, and lifespan (Fig 7). *lipl-1* enhances immune resistance in *tcer-1* mutants, whereas, *lipl-2* does not. Instead, *lipl-2* appears to limit longevity and immunity especially upon *tcer-1 lof*. Both lipases support fertility, especially embryonic integrity under infection stress, with *lipl-2* playing a more prominent role. Their lipidomic impacts are characterized by shared features well as production of specific lipids found to be correlated with stress resistance, lifespan and healthspan. Amongst these, we identified Cer 24 as a LIPL-1-derived species that promotes immunity in infected animals. Further, we found that the human ortholog, LAL/LIPA, rescues *lipl-1*-associated immune defects and improves survival upon infection suggesting potential evolutionary conservation of these molecular functions.

A particularly revealing aspect of our study is the alteration of specific, functionally relevant sphingolipid species in a lipase-dependent manner. LIPL-1 and LIPL-2 did not broadly alter global neutral lipid or phospholipid levels but instead modulated select species, notably ceramide sphingolipids. This aligns with growing evidence that overall fat or TAG levels are poor indicators of lifespan or stress resilience: long-lived insulin/IGF-1 mutants such as *daf-2* and germline-ablated animals accumulate fat, whereas calorie-restricted animals are lean yet long-lived [63,64]. Here, we found that *tcer-1* mutants accumulated bioactive ceramides Cer 22, Cer 24 and Cer 26 dependent on *lipl-1* activity. Ceramides are essential membrane lipids with critical signaling roles [65,66], and distinct species in *C. elegans* have been linked to mitochondrial surveillance (Cer 24), heat-stress resistance (Cer 16), and germline ablation- induced longevity (GlcCer C22) [67–69]. The specificity of ceramide changes likely reflects the influence of their acyl chain composition, which affects lipid shape, protein interactions, and function. While alterations in ceramides associated with stress and longevity have previously been attributed to regulation of biosynthetic enzymes such as the ceramide synthase *hyl-2*, our RNA-seq data did

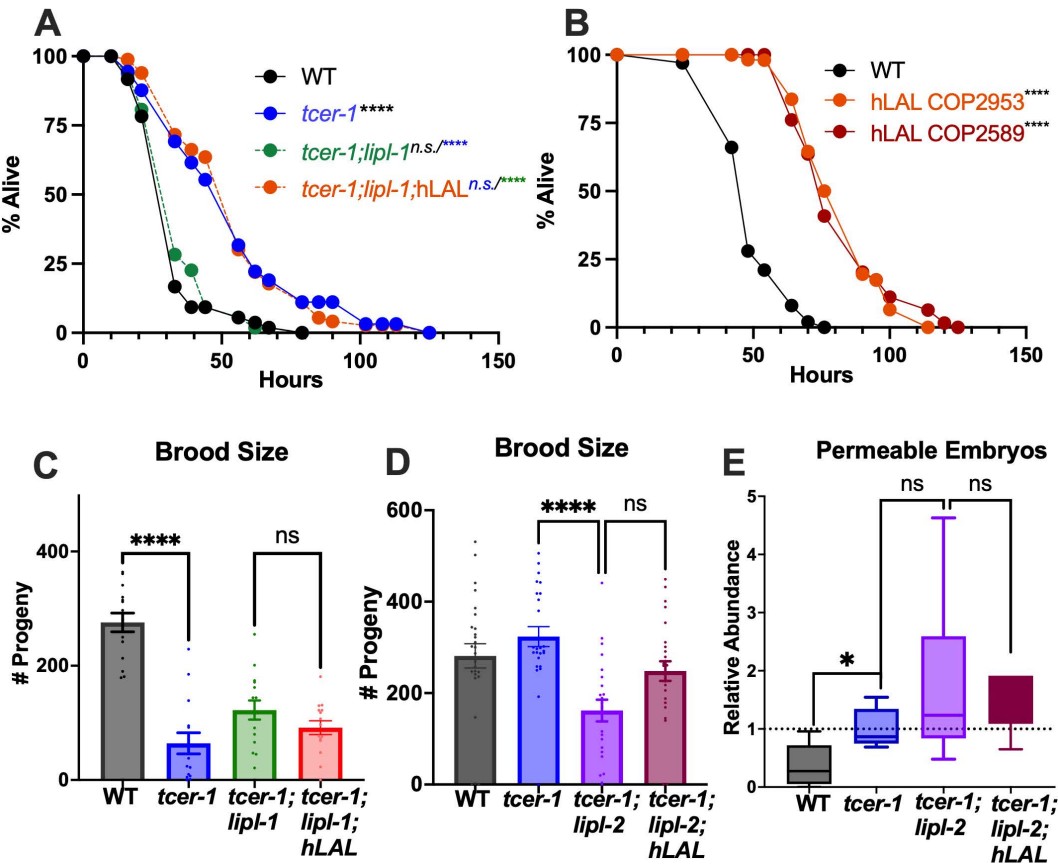

**Fig 6. Human Lysosomal Acid Lipase (hLAL/LIPA) rescues immune deficits induced by *lipl-1* mutation and enhances survival upon PA14 infection. A: hLAL/LIPA rescues survival of *tcer-1;lipl-1* mutants on PA14 exposure to *tcer-1* level.** Survival upon PA14 exposure from late-L4 stage onwards compared between wild type WT (black, m = 35.06 ± 1.91, n = 54/90), *tcer-1* (blue, m = 54.8 ± 3.25, n = 63/90), *tcer-1;lipl-1* (green, m = 36.65 ± 1.79, n = 52/90), *tcer-1;lipl-1;hLAL* (red, m = 55.77 ± 2.53, n = 73/90). **B: hLAL/LIPA expression in *C. elegans* enhances survival upon PA14 infection.** Survival upon PA14 exposure from late-L4 stage onwards compared between WT (black, m = 57.24 ± 2.01, n = 35/110) and two transgenic strains expressing hLAL broadly in somatic tissues. COP2983 (orange, m = 83.18 ± 2.22, n = 46/96) and COP2589 (red, m = 82.84 ± 2.22, n = 63/103). Data from additional trials in S8 Table. **C-E: hLAL/LIPA does not rescue the reproductive defects of *tcer-1;lipl-1* and *tcer-1;lipl-2* mutants.** Comparisons of reproductive success in wild type (WT, black), *tcer-1;lipl-1* (green), *tcer-1;lipl-2* (purple), *tcer-1;lipl-1;hLAL* (red) and *tcer-1;lipl-2;hLAL* (maroon) strains. **C, D:** Brood size: Total number of live progeny counted per worm per strain. Each dot represents one worm from an aggregate for two independent trials. **C:** WT (n = 15, m = 275.7 ± 16.37), *tcer-1* (n = 15, m = 64.20 ± 18.40), *tcer-1;lipl-1* (n = 15, m = 122.4 ± 6.80), *tcer-1;lipl-1;hLAL* (n = 15, m = 91.53 ± 11.99). **D:** WT (n = 26, m = 281.5 ± 26.43), *tcer-1* (n = 25, m = 323.6 ± 21.83), *tcer-1;lipl-2* (n = 23, m = 161.9 ± 23.66), *tcer-1;lipl-2;hLAL* (n = 26, m = 248.3 ± 21.53). **E:** Quantification of Embryonic egg-shell defects. Average-fold change of the fraction of BODIPY stained embryos laid by Day 4 mothers, normalized to *tcer-1* mutants. WT (n = 126, m = 0.3687 ± 0.1515), *tcer-1* (n = 30, m = 1.0 ± 0.1389), *tcer-1;lipl-2* (n = 45, m = 1.741 ± 0.6080), *tcer-1;lipl-2;hLAL* (n = 36, m = 1.459 ± 0.2039). In A-B, statistical significance was calculated using the log-rank Mantel Cox method and is shown on each panel next to a given strain/condition with the color of the asterisk indicating strain being used for comparison. In C-D, statistical significance was calculated using one-way ANOVA with Tukey's correction. In E, Student's t-test was used. p ≤ 0.05(*), *p* < 0.01 (**), < 0.001 (***), < 0.0001(****), n.s: not significant.

not show the differential expression of *hyl-2* or other sphingolipid biogenesis genes following pathogen infection or *tcer-1* loss. Hence, these observed immune-related ceramide changes appear lipolysis-driven, consistent with ceramide generation by lysosomal lipases through breakdown of complex sphingolipids in mammals [70]. Importantly, Cer 24 supplementation restored *tcer-1;lipl-1* mutants' survival on PA14 and improved post-infection resilience in WT animals, underscoring the physiological significance of such specific lipid species. Analogously, in humans, distinct ceramides rather than total lipid content determine metabolic outcomes. For instance, elevated C16:0 and C18:0 ceramides correlate with insulin

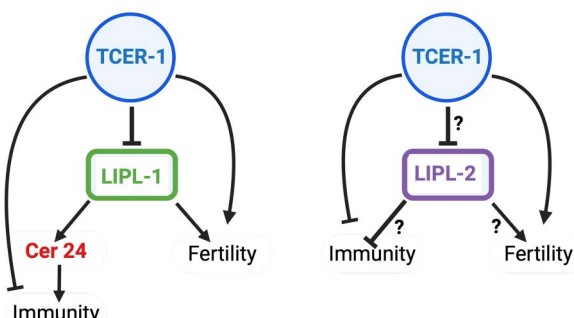

**Fig 7. Schematic depicting the regulation of lysosomal lipases, *lipl-1* and *lipl-2*, by TCER-1 and their distinct roles in immunity and fertility identified in this study.** TCER-1 represses the expression of *lipl-1* upon infection, which in turn modulates levels of the specific fatty-acid species, such as the Ceramide 24 (Cer 24) that promote immune resilience. *lipl-1* also acts with TCER-1 to promote fertility (left panel). In contrast, the relationship between TCER-1 and *lipl-2* appears more complex. Although TCER-1 represses *lipl-2* induction upon infection, both genes repress immunity and promote fertility, suggesting that TCER-1 may act through additional pathways to coordinate these processes (right panel). Arrows denote activation, and blunt-ended lines denote repression. Figure created using Biorender.

resistance and β-cell dysfunction in Type 2 Diabetes [71,72], whereas, exercise selectively reduces C16:0 and enriches very-long-chain ceramides (C24:0, C24:1), enhancing insulin sensitivity and mitochondrial capacity [73]. Together, these findings highlight that targeted remodeling of ceramide composition, not total fat content, underlies adaptive and pathological metabolic states.

An unexpected finding of our study is the contrasting roles of *lipl-1* and *lipl-2* in immune defense. Our RNAseq data had led us to hypothesize that TCER-1 suppresses immunity by downregulating both *lipl-1* and *lipl-2* and we had thus envisaged a pro-immunity function for both genes. While *tcer-1* mutants did show *lipl-1* and *lipl-2* transcriptional upregulation during PA14 infection, relative to uninfected animals, only *lipl-1* was required for the enhanced pathogen resistance of *tcer-1* mutants, not *lipl-2*. In fact, *lipl-2* loss appeared to enhance post-infection survival and longevity in *tcer-1* mutants, highlighting that transcriptional changes do not always reflect functional impact. We also observed discrepancies between *in vivo* mRNA and protein reporters, with low mRNA and very low protein in adult tissues, suggesting tight post-transcriptional control and rapid protein turnover. The lipases' protein reporters localized to coelomocytes despite intestinal transcription, implying secretion- consistent with prior findings on LIPL-1, -3, and -5 [37,38]. Temperature also played a role: although *tcer-1 lof* consistently increased reporter expression in all conditions- matching gene expression data- the 25°C temperature required for PA14 virulence elevated reporter expression too, likely reflecting previously-described HSF-1- dependent heat shock response [74].

*lipl-2* supports reproductive health, particularly eggshell integrity during infection. Strikingly, the combined loss of *tcer-1* and *lipl-2* caused even young mothers to lay defective embryos. These phenotypes are noteworthy considering the unique lipidomic changes we observed in *tcer-1;lipl-2* mutants. The abundance of 7 lipid species was altered in *tcer-1;lipl-2* mutants compared to *tcer-1* alone. Of these, elevation of three species- LPC 18:2, PC 32:1, and PC 32:2- has been linked to adverse pregnancy outcomes in women [75,76]. Additionally, LPC 18:2 and PC 32:3, also elevated in *tcer-1;lipl-2*, are part of a panel of 8 serum biomarkers linked to human healthspan. In elderly individuals, higher LPC 18:2 levels predict slower decline in gait speed, while PC 32:3 correlates with reduced gait speed [60].

The ability of the hLAL/LIPA to substitute for *lipl-1* in promoting immune resistance is promising because of its known roles in macrophage biology, wherein, it regulates inflammation and is essential for M2 macrophage polarization, which supports anti-inflammatory responses and tissue repair [61]. Its deficiency causes lipid storage disorders such as Wolman's disease and cholesterol ester storage disease (CESD) and leads to tissue-specific inflammation [35]. In mice, LAL/LIPA deficiency leads to aberrant macrophage infiltration, while tissue-specific knockouts in the liver, intestine, and lung

exhibit heightened inflammation in the affected tissues [77–79]. In our experiments, hLAL/LIPA did not produce a significant or reliable rescue of *lipl-1* or *lipl-2 lof* fertility phenotypes. Although the effects of *lipl-2* are complex, the lipidomic parallels between *C. elegans* LPC changes and human health metrics are compelling. Together with the rescue of *lipl-1 lof* immunity phenotypes by hLAL/LIPA, they raise the enticing possibility that these genes may play conserved roles in the immune-reproductive axis.

## Materials and methods

### *C. elegans* and bacterial strains

All strains were maintained by standard techniques at 20°C or 15°C on nematode growth medium (NGM) plates seeded with an *E. coli* strain OP50. For experiments involving RNAi, NGM plates were supplemented with 1 mL 100 mg/mL Ampicillin and 1 mL 1M IPTG (Isopropyl β-D-1-thiogalactopyranoside) per liter of NGM. The main strains used in this study are listed in S9 Table.

### Generation of *lipl-1* and *lipl-2* reporter strains

*Translational Reporters:* The *lipl-2p*::LIPL-2::RFP construct was a gift from Dr. Abhinav Diwan [39]. *lipl-1p*::LIPL-1::RFP was generated by amplifying 1,010 bp of the endogenous *lipl-1* promoter and entire CDS using primers which introduced Xma1 and Acc65i cut sites and eliminated the CDS stop codon (S10 Table). This was cloned into the *lipl-2p*::LIPL-2::RFP in PDG219 construct using restriction enzyme cloning to replace the *lipl-2* sequence. Transgenic strains were generated by injecting reporter constructs (25ng/µl) with a co-injection marker containing *myo-2p*:GFP (15ng/µl) Strains were maintained by picking green fluorescent worms. *Transcriptional reporters:* For each construct, the selected region of the *lipl-1* (441 bp or 994 bp) or *lipl-2* (1000 bp or 1500 bp) endogenous promoter was amplified using primers which added a homology region to the pAV1944 construct containing *myo-2p*:mCherry. pAV1944 was linearized with Nhe1 and Sph1 restriction enzyme digestion to excise the *myo-2* promoter, and the final constructs were assembled using a Gibson assembly kit (NEB E5520S) according to manufacturer directions. Transgenic strains were generated by injecting a *lipl-* reporter construct (50ng/µl) along with a co-injection marker (*ofm-1p*:GFP or *myo-2p*:GFP, 15ng/µl). Strains were maintained by picking green fluorescent worms. The translational reporter strains showed developmental variability and propensity for loss of transgenes across generations which can likely be attributed to their array-based construction. Primers used in this study are listed in S10 Table.

### Generation of *lipl-1* and *lipl-2* deletion mutants

Strains were generated using the co-CRISPR method previously described [80]. Briefly, gRNAs and repair templates were designed to excise the entire CDS of *lipl-1* or *lipl-2* (S10 Table). Injection mix (10µl volume: 24µg Cas9, 20µg tracrRNA (IDT 1072533), 4µg each target gRNA, 500ng target repair template, 2µg *dpy-10* crRNA, 250ng *dpy-10* repair template, IDT Nuclease-free Duplex Buffer to 10µL) was freshly prepared and spun for 20 minutes at 12,000g. Young Day 1 adults were injected in the gonad, and monitored for dumpy-roller progeny. F1 larvae from *dpy/rol* plates were screened by PCR to identify heterozygous CRISPR deletion alleles, which were outcrossed to N2 thrice and homozygosed. Primers, gRNAs and repair templates used are listed in S10 Table.

### Generation of hLAL strains

The transgenic strains COP2589 {*knuSi924 [pNU3447 (eft-3p::hLIPA::linker::wrmScarlet::3xFLAG::tbb-2u in cxTi10882, unc-119(+)) IV; unc-119(ed3) III*} and COP2593 {*knuSi927 [pNU3447 (eft-3p::hLIPA::linker::wrmScarlet::3xFLAG::tbb-2u in cxTi10882, unc-119(+))} IV; unc-119(ed3) III*} expressing hLAL/LIPA using the pan-somatic promoter, *eft-3*, were created by InVivo Biosystems using the Mos1-mediated Singly Copy Insertions (MosSCI) method which enables integration of

a transgene as a single copy at a designated *C. elegans* locus [81]. The *unc-119* rescue cassette was used to bring the transgene into a target Mos1 locus on chromosome IV and create rescue of the *unc-119(ed3)* mutant allele. The Mos1 locus was selected for position-neutral effects and to avoid the gene-coding regions, introns and transcription factor binding sites. Transgene integration was confirmed by PCR.

### Fluorescence imaging and quantification

*Transcriptional Reporters:* Transgenic animals were immobilized with 20 mM Levamisole, mounted on agar pads and imaged using a Leica DM5500B compound scope at 20x. Image acquisition was performed using LAS X software (Leica). Images were processed using ImageJ software. *Translational Reporters:* Transgenic animals were immobilized with 20 mM Levamisole, mounted on agar pads and imaged using a Leica Stellaris 5 confocal with an integrated White Light Laser (WLL) at 20x and 63x. Image acquisition was performed using LAS X software (Leica). Images were processed using ImageJ software. Fluorescence intensity was quantified using the Lookup Table (LUT) 'Fire' feature applied to pseudocolor images. The LUT Fire gradient mapped pixel intensity from low to intermediate (red/yellow) to high (white) fluorescence, allowing visual discrimination of signal intensity across samples, differentiation between regions of comparable size but differing fluorescence levels and delineating true signal from background. Quantification was performed by measuring mean pixel intensity within defined regions of interest (ROIs) on identically processed images.

### Lifespan assays

Lifespan experiments were performed as previously described [82]. All lifespan experiments were conducted at 20 °C on *E. coli* OP50 plates unless otherwise noted. Between 10–15 L4 hermaphrodites were transferred to each of ~5−6 plates per experiment and observed at 24−48 h intervals to document live, dead or censored (animals that exploded, bagged or could not be located) animals. Animals were scored as dead when they failed to respond to gentle prodding with a platinum wire pick. Fertile strains were transferred every other day to fresh plates until progeny production ceased. The program Online Application of Survival Analysis 2 (OASIS 2) [83] was used for statistical analysis of both lifespan and pathogen stress assays. P-values were calculated using the log-rank (Mantel–Cox) test and subjected to multiplicity correction in experiments that involved more than two strains/conditions. Results were graphed using GraphPad Prism (Version 8).

### PA14-infection survival assays

Pathogenic bacterial strain *Pseudomonas aeruginosa* (strain PA14) was streaked from frozen stocks onto Luria Bertani (LB) agar, incubated at 37°C overnight and stored at 4°C for a week or less. Single colonies from the streaked plates were inoculated and grown in King's Broth to exponential growth phase, 6–12 hours at 37 °C with shaking. ~20 µl of this broth culture was seeded onto slow killing (SK) plates (modified NGM plates containing 0.35% peptone instead of 0.25%) and incubated for 24 h at 37 °C. The plates were then left to sit at room temperature (RT) for 24 h prior to use. Between 30 and 40 L4 hermaphrodites per strain were transferred to each of ~3–5 PA14 plates, incubated at 25 °C and monitored at 6−12 h intervals to account for live, dead or censored animals as described above. To rule out the impact of internal hatching on experimental outcomes, L4 larval stage animals were treated with 100 µg per ml of Fluoro Deoxy Uridine (fUDR) on NGM plates with OP50. Exposing *C. elegans* to this treatment for 24 h at $20^0$C before transferring to PA14 SK plates prevented the eggs from hatching, without significant impact on the strains' survival dynamics (S7 Table). For experiments that also involved RNAi treatment, animals were grown to the L4 stage on standard RNAi plates seeded with *E. coli* HT115 strain carrying an empty vector control (pAD12 or L4440) or the relevant RNAi clone before transferring to PA14-seeded SK plates and assaying for survival at 25 °C. Kaplan−Meier analysis and statistics were performed as described above for lifespan assays.

## Lipid supplementation assays

Cer 24 ceramide (Avanti Polar Lipids, Cat. No. 860649P) and lysophosphatidylcholine 18:2 (LPC 18:2; MedChemExpress, Cat. No. HY-N9410) were dissolved in ethanol to prepare stock solutions at a concentration of 0.5 mg/mL. For supplementation, 60µg of Cer 24 or LPC 18:2 (120 µL of the stock solution) was evenly applied to the surface of NGM plates prior to bacterial seeding. Control plates received an equal volume of ethanol (120 µL). L4-stage worms were transferred to either control or Cer 24 plates prior to infection, and Day 1 adults were subsequently exposed to the pathogen. Pathogen exposed worms were transferred daily to freshly prepared control or Cer 24 or LPC 18:2 plates.

## Reproductive-health assay

Reproductive health was assessed using previously described methods [33]. All experiments were conducted at 20°C and when matricide/ bagging occurred the animal was censored from the experiment on that day. Individual synchronized L4 hermaphrodites were moved to fresh plates daily till the end of the reproductive phase (no progeny observed on a plate for a minimum of 2 days). Each day, once the parent was moved to a fresh plate, the older plate with eggs was stored at 20°C for ~48 hours, and the number of hatched worms and eggs counted to calculate brood size. The above two parameters were used to determine viability (ratio of the total number of eggs laid by a hermaphrodite in its lifetime to total number of eggs that hatched). Animals that failed to lay any viable embryos were defined as sterile and this fraction was used to estimate sterility in the population. The data presented was obtained from aggregation of two to three independent trials wherein at least 10–15 animals per strain were examined.

## Embryo permeability assessment

For each genotype, approximately 20 gravid mothers were allowed to lay eggs overnight on 6 cm plates. After 12hrs, the mothers were removed, and 200µl working solution of 20µg/mL BODIPY (Invitrogen D3922) in Egg Buffer [4mM HEPES (pH 7.4), 94mM NaCl, 32 mM KCl, 2.7mM $CaCl_2$ and 2.7mM $MgCl_2$] was added to flood each plate. After the liquid was completely absorbed, embryos were immediately counted and screened for fluorescence using a Leica DM3000B microscope with a Lumen 200 metal halide fluorescence illuminator light box (Prior Scientific L200). For Day 4 experiments, the worms were transferred to fresh plates every day till Day 4 and then allowed to lay eggs before the same protocol was used.

## RNA-sequencing and data analysis

RNA was isolated from 3 biological replicates of age-synchronized, Day 1 WT animals and CF2166 *tcer-1(tm1452)* mutants grown on OP50 till late L4 stage (~48h), then transferred to PA14 plates (grown as described above) or allowed to continue on OP50. After 8h of exposure, worms were harvested for RNA isolation from approximately 3000 worms per strain. Following 7 freeze thaw cycles, RNA was isolated using the Trizol method and for quality and quantity using the Agilent Tape station and Qubit Fluorometry. Sequencing libraries were prepared using the TruSeq stranded mRNA (PolyA+) kit and the samples were then subjected to 75 base pair paired-end sequencing on an Illumina NextSeq 500 sequencer at the Univ. of Pittsburgh Genomics Research Core. Sequencing data was analyzed using the CLC Genomics Workbench (Version 20.0.3) employing the RNA Seq pipeline. Differentially regulated genes were filtered for significant changes based on the criteria of >2 fold change in expression, P Value of <0.05 and a false discovery rate (FDR) of <0.05.

## Gene ontology analyses

Genes that were differentially regulated in a statistically significant manner were classified into two groups as either up-regulated (UP) or down-regulated (DOWN) targets. These groups were analyzed for enrichment of gene classes based on Gene Ontology (GO) Terms using *C. elegans* centered publicly available online resources, Wormbase Gene

Set Enrichment Analysis tool (https://wormbase.org/tools/enrichment/tea/tea.cgi) and WormCat (http://wormcat.com/) [84]. Representation Factor was calculated at http://nemates.org/MA/progs/overlap_stats.html.

## Lipidomics: Sample collection

From each strain, ~5,000 young Day 1 adults were collected at approximate onset of egg lay. To obtain these, approximately 600 gravid mothers were placed on 10 cm plates seeded with OP50 and allowed to lay eggs for 5–8 hours, then moved to fresh plates to lay eggs overnight. The next day, all mothers were removed from overnight egg plates, and plates were subsequently monitored for onset of egg lay. WT, *lipl-1*, and *lipl-2* young adults were collected by washing plates with M9. *tcer-1*, *tcer-1;lipl-1*, and *tcer-1;lipl-2* strains exhibit delayed, unsynchronized development and significant infertility. To ensure that only fertile young adults were collected in these strains, once the population had abundant L4s, about 5,000 L4s were moved to a fresh 10 cm plate. The next day, only individuals with eggs were manually picked for the sample collection. Due to the labor-intensive nature of this approach, the sample collection was performed in two batches with WT and *tcer-1* controls included in each batch: (i) WT, *lipl-1*, *tcer-1*, *tcer-1;lipl-1* and (ii) WT, *lipl-2*, *tcer-1*, *tcer-1;lipl-2.* Five or six independent biological replicates were collected in each batch for each strain. Upon collection, worms were rinsed 2x with M9 and centrifugation (1000g x 2min). Final pellet was rinsed 1x with Millipore water and flash frozen.

## Lipid extraction

Lipid extraction and detection of PL, NL and sphingolipids was carried out as described [85]. Briefly, for PLs, total lipids were extracted from nematodes strain using a chloroform/methanol (2:1 v/v) solvent system. A 1,2-diundecanoyl-sn-glyerco-3- phosphocholine standard was added for relative quantification prior to extraction (Avanti Polar Lipids). Total lipids were injected onto the LC-MS/MS system for the negative ion scanning mode analysis using an HPLC system (Dionex UHPLC UltiMate 3000) equipped with a C18 Hypersil Gold 2.1×50 mm, 1.9 μm column (25002–052130; Thermo Scientific) equipped with a 2.1 mm ID, 5 μm Drop-In guard cartridge (25005–012101; Thermo Scientific). Analysis was completed using a Q Exactive Orbitrap mass spectrometer coupled with a heated electrospray ionization source (Thermo Scientific). For extraction and detection of NLs, the same protocol was followed with minor changes. The standard for relative analysis was a triglyceride standard mix (GLC-406 Nu-Chek). Lipids were injected onto the HPLC-MS/MS system for the positive ion scanning mode analysis. For sphingolipid extraction and detection, a Ceramide/ Sphingoid Internal Standard Mixture I (Avanti Polar Lipids) was added to the nematode sample and total lipids extracted as previously mentioned using a chloroform:methanol mixture. Sphingolipids were separated and analyzed using the same instrument as PLs and NLs with the following modifications. 10 μl of the purified sphingolipids were injected onto the HPLC-MS/MS system for the positive ion scanning mode analysis. For MS analysis, the following parameters were used: the capillary temperature was set at 275 °C, the sheath gas flow rate was set at 45 units, the auxiliary gas flow was set at 10 units, the source voltage was 3.2 kV, and the AGC target was 106. Acquisition was carried out with full-scan data-dependent MS2 (ddMS2) mode. For MS1 profiling, scans were run at a resolution of 70k. MS2 analyses were performed using five scan events, where the top five ions were chosen from an initial MS1 scan. For fragmentation, a normalized collision energy of 50 was used. MS1 spectra were collected in profile mode, whereas MS2 spectra were collected in centroid mode. Overall, for the strains *lipl-1, lipl-2, tcer-1; lipl-1* and *tcer-1; lipl-2,* between 4–6 successful MS runs were conducted and for WT and *tcer-1* data obtained from 7 to 10 successful MS runs was used for subsequent analysis.

## Data analysis

Lipid analysis of the LC-MS/MS data was carried out using the software Lipid Data Analyzer (LDA) Version 2.8.1. LDA mass lists were generated for PLs and sphingolipids based on our previous studies [86]. LDA mass lists for NLs were created for all the neutral headgroups listed on LIPID MAPS. LDA exact mass lists were generated for diacylglycerols (DG), betaine diradylglycerols (DGTA), monogalactosyldiacylglycerol (MGDG), triacylglycerol (TG), alkyldiacylglycerols (TGO),

and glycosyldiacylglycerols (HEXDG) using the following ranges: the molecular species with combined chain lengths of 24–44, 28–40, 24–38, 24–66, 37, and 45 respectively and degrees of unsaturation of 0–12 for all classes. Spectra from MS2 scans were used to validate LDA species identifications, which was also compared to LIPID MAPS product ion prediction tools. A relative quantification was used to compare between samples as it allowed us to internally control for sample size of thousands of worms while allowing for detection of fluctuations in less abundant species and produce reliable results.

## Supporting information

**S1 Fig. Overlap between PA14-induced differentially expressed genes (DEGs) identified in this study and previously reported PA14-induced gene lists. A:** All genes up- and down-regulated upon WT animals' exposure to PA14 for 8 hours in this study compared with genes identified by Troemel et al. after 4 hours and 8 hours of exposure [27]. **B-E:** Comparisons of DEGs upregulated **(B, D)** or downregulated **(C, E)** on PA14 with genes identified by Shapira et al., 2006 **(B, C)** [26] and Twumasi-Boateng et al. 2012 **(D, E)** [47]. RF: Representation Factor. Statistical significance of overlap between gene sets calculated using hypergeometric probability formula with normal approximation (see Methods).
(TIF)

**S2 Fig. Global and category-specific transcriptional alterations induced by PA14 infection in WT and *tcer-1* mutants identified using WormCat [48].** Analysis of gene expression changes in WT animals and *tcer-1* mutants upon PA14 infection. Panels A and B show the global transcriptional response driven by infection (A) and genotype (B), while panels C-H detail category-specific changes in *tcer-1* mutants on PA14 compared to WT animals on PA14. **A:** Bubble plot of genes upregulated (UP) or downregulated (right) in WT (N2) and *tcer-1* mutants following PA14 exposure compared to OP50. Results of Categories 1, 2 and 3 are shown. **B:** Comparison of genes up- or downregulated in *tcer-1* mutants relative to WT on OP50 or on PA14 infection. Upregulated genes (UP) and downregulated genes (right) are plotted across three categories as in A. **C-H:** Detailed breakdown of genes upregulated (C-E) or downregulated (F-H) in *tcer-1* mutants versus WT under PA14 infection across three hierarchical categories, 1 (C, F), 2 (D, G) and 3 (E, H). Bubble size represents the number of genes in each WormCat category, and color intensity reflects statistical significance.
(TIF)

**S3 Fig. Categorical analysis of *lipl-1p::mCherry* expression.** Fluorescence levels quantified based on area and observable intensity variation (see Methods). **A, B: Intestinal Expression. A:** Representative images of each category pseudocolored with ImageJ LUT Fire. ***Low-*** dim fluorescence primarily visible in posterior and anterior intestine. No areas of high intensity; posterior fluorescence limited to region near tail. ***Medium-*** posterior fluorescence between vulva and tail. Some areas of intermediate intensity in anterior intestine. ***High-*** posterior fluorescence uniformly extended up to the vulva or with multiple areas of bright intensity. ***Very High–*** posterior intestine with very bright fluorescence that extends into anterior half beyond the vulva and intermediate fluorescence extended at least to the vulva. Purple boxes indicate categories quantified in Fig 2E, 2F. **B:** Quantification of percent of population in each category. Data from 3 pooled biological replicates. EV: Empty vector control. (ev RNAi till L4 then OP50, 20°C, n = 81), (ev RNAi till L4 then PA14 20°C, n = 62), (*tcer-1* RNAi till L4 then OP50, 20°C, n = 83), *(tcer-1* RNAi till L4 then PA14 20°C, n = 41), (ev RNAi till L4 then OP50, 25°C, n = 82), (ev RNAi till L4 then PA14, 25°C, n = 69), (*tcer-1* RNAi till L4 then OP50, 25°C, n = 84), *(tcer-1* RNAi till L4 then PA14, 25°C, n = 68) **C, D: Expression in Head Region. C:** Representative images of each category pseudocolored with ImageJ LUT Fire. ***Low-*** No fluorescence visible. ***Medium-*** 1 or 2 puncta seen. ***High-*** More than 2 puncta, or diffuse, non-punctate expression. **D:** Quantification of data from 3 pooled biological replicates. (ev RNAi till L4 then OP50 20°C, n = 81), (ev RNAi till L4 then PA14 20°C, n = 62), (*tcer-1* RNAi till L4 then OP50 20°C, n = 83), *(tcer-1* RNAi till L4 then PA14

20°C, n = 41), (ev RNAi till L4 then OP50 25°C, n = 82), (ev RNAi till L4 then PA14 25°C, n = 69), (*tcer-1* RNAi OP50 25°C, n = 84), *(tcer-1* RNAi PA14 25°C, n = 67).
(TIF)

**S4 Fig. LIPL-1 and LIPL-2 protein levels observed using translational reporter strains *lipl-1p*::LIPL-1::RFP and *lipl-2p*::LIPL-2::RFP.** A-D and E-H show representative images of Day 1 adults with *lipl-1p*::LIPL-1::RFP and *lipl-2p*::LIPL-2::RFP expression, respectively, that was predominantly seen in coelomocytes. Animals were raised on bacteria expressing control empty vector (Ctrl) or *tcer-1* dsRNA (RNAi) until young adulthood, transferred to plates seeded with PA14 or OP50 bacteria and incubated for 8 hours at 25 °C. Images pseudocolored with ImageJ LUT RedHOT showing very low expression predominantly in coelomocytes (with high intestinal autofluorescence in E-H).
(TIF)

**S5 Fig. Categorical analysis of *lipl-2p::mCherry* expression.** Expression levels in a population were variable and primarily observed in intestines. Fluorescence quantified based on area and observable intensity variation (see Methods). **A, B: Intestinal Expression. A:** Representative images of each category pseudocolored with ImageJ LUT Fire. ***Low-*** dim fluorescence primarily visible in posterior intestine limited to region between vulva and tail. Brighter fluorescence, if present, was restricted to tail. ***Medium-*** medium fluorescence in region between vulva and tail. Some area of increased intensity which extended past tail into anterior regions. ***High*** – broader posterior signal extending beyond vulva to anterior intestine with areas of bright intensity in posterior intestine. ***Very High*** – Intense fluorescence extended from posterior intestine to anterior of vulva. Purple boxes indicate categories quantified in elevated expression analysis in Fig 2K and 2L. **B:** Quantification of percent of population in each category. Data from 3 pooled biological replicates. EV: Empty vector control. (ev RNAi till L4 then OP50 20°C, n = 86), (ev RNAi till L4 then PA14 20°C, n = 81), (*tcer-1* RNAi till L4 then OP50 20°C, n = 83), *(tcer-1* RNAi PA14 till L4 then 25°C, n = 78), (ev RNAi till L4 then OP50 25°C, n = 87), (ev RNAi till L4 then PA14 25°C, n = 81), (*tcer-1* RNAi till L4 then OP50 25°C, n = 90), *(tcer-1* RNAi till L4 then PA14 25°C, n = 72) **C: Expression dynamics across lifespan.** Representative images of *lipl-2p*:mCherry expression in (from left to right) young larvae, L4 larvae, newly-hatched young adult, Day 1 adult, and Day 5 adult worms. Pseudocolored in ImageJ with LUT Fire.
(TIF)

**S6 Fig. Loss of *tcer-1*, *lipl-1* and *lipl-2* causes embryonic eggshell defects with age.** Longitudinal analysis of BODIPY-permeable eggs laid by Day 1 (D1) and Day 4 (D4) mothers by different strains. WT (D1 0, n = 712; D4 0, n = 276), *lipl-l* (D1 0, n = 700; D4 0.4267, n = 703), *lipl-2* (D1 0.0356, n = 2806; D4 0.9451, n = 529), *tcer-1* (D1 0, n = 896; D4 0.3361, n = 596), *tcer-1;lipl-1* (D1 0, n = 1456; D4 0.4687, n = 640), *tcer-1;lipl-2* (D1 0.2212, n = 1356; D4 1.594, n = 439).
(TIFF)

**S7 Fig. Impact of *lipl-1* and *lipl-2* deletions on relative abundance of neutral lipid (NL) and phospholipid (PL) classes.** Lipids from gravid Day 1 adults were analyzed using HPLC-MS/MS. **A, B:** Impact of *lipl-1* (A) or *lipl-2* (B) inactivation on relative abundance of NLs with > 2% abundance in total NL population. **C, D:** Relative abundance of PL categories altered by *lipl-1* (C) or *lipl-2* (D) inactivation. Statistical significance was calculated using two-way ANOVA with Tukey's correction, $p \leq 0.05(*)$, $p < 0.01$ (**), $< 0.001$ (***), $< 0.0001$ (****).
(TIFF)

**S8 Fig. Impact of *lipl-1* and *lipl-2* deletions on saturation and fatty-acid chain length in neutral lipids. A, B:** Relative abundance of double bonds (DB) in triacylglycerides (TAGs) altered by *lipl-1* (A) or *lipl-2* (B) inactivation. **C, D:** Relative abundance fatty-acid chain length (CL) of TAGs altered by *lipl-1* (C) or *lipl-2* (D) inactivation. Color key indicated on each panel. Statistical significance was calculated using two-way ANOVA with Tukey's correction, $p \leq 0.05(*)$, $p < 0.01$ (**), $< 0.001$ (***), $< 0.0001$ (****).
(TIFF)

**S9 Fig. Impact of *lipl-1* and *lipl-2* deletions on saturation and fatty-acid chain length of phosphatidylcholine (PC) and phosphatidyl-ethanolamine (PE) lipids. A, B:** Relative abundance of PC double bonds (DB) altered by *lipl-1* (A) or *lipl-2* (B) inactivation. **C, D:** Relative abundance of PC chain length (CL) altered by *lipl-1* (C) or *lipl-2* (D) inactivation. **E, F:** Relative abundance of PE double bonds altered by *lipl-1* (E) or *lipl-2* (F) inactivation. **G, H:** Relative abundance of PE chain length altered by *lipl-1* (G) or *lipl-2* (H) inactivation. Color key of different strains shown at top. Statistical significance was calculated using two-way ANOVA with Tukey's correction, $p \le 0.05$(*), $p < 0.01$ (**), $< 0.001$ (***), $< 0.0001$ (****).
(TIFF)

**S10 Fig. Impact of *lipl-1* and *lipl-2* deletions on glucosyl-ceramides (GlcCer) and ceramides (Cer) lipids. A, B:** Relative abundance of Cers altered by *lipl-1* (A) or *lipl-2* (B) inactivation. **C, D:** Relative Abundance of GlcCers altered by *lipl-1* (C) or *lipl-2* (D) inactivation. Color key of different strains shown at top. Statistical significance was calculated using two-way ANOVA with Tukey's correction, $p \le 0.05$ (*), $p < 0.01$ (**), $< 0.001$ (***), $< 0.0001$ (****).
(TIFF)

**S1 Table. Gene lists and GO-term analyses of differentially expressed genes identified by RNAseq in this study.**
(XLSX)

**S2 Table. Overlaps between PA14-induced genes identified in this study and previously reported PA14-driven genes.**
(XLSX)

**S3 Table. Survival of different strains upon PA14 infection.**
(PDF)

**S4 Table. Impact of *lipl-1* and *lipl-2* null mutants on lifespan on *E. coli* OP50.**
(PDF)

**S5 Table. Lipid species altered in *tcer-1* mutants and the impacts of *lipl-1* and *lipl-2* mutations on these alterations.**
(PDF)

**S6 Table. Lipid species altered in *lipl-1* and *lipl-2* single mutants.**
(PDF)

**S7 Table. Survival of Cer 24 and LPC 18:2 supplemented strains upon PA14 infection.**
(PDF)

**S8 Table. Survival of strains expressing human LAL (hLAL/LIPA) in different genetic backgrounds on PA14.**
(PDF)

**S9 Table. Strains used in this study.**
(PDF)

**S10 Table. Primers used in this study.**
(PDF)

## Acknowledgments

The authors are grateful to members of the Ghazi lab and the Pittsburgh 'Wormclub' community for valuable inputs throughout this study. The intellectual and experimental inputs provided by Javier Irazoqui and Jiali Shen (UMass Chan Medical School) is gratefully acknowledged. Thanks to Ho Yi-Mak (HKUST), John Murphy and Abhinav Diwan

(Washington University) for generously sharing strains. Some strains were provided by the CGC, which is funded by NIH Office of Research Infrastructure Programs (P40 OD010440).

## Author contributions

**Conceptualization:** Arjumand Ghazi.

**Data curation:** Laura Bahr, Francis RG Amrit, Paige Emily Silvia, Carissa Perez Olsen, Arjumand Ghazi.

**Formal analysis:** Laura Bahr, Francis RG Amrit, Paige Emily Silvia, Carissa Perez Olsen, Arjumand Ghazi.

**Funding acquisition:** Arjumand Ghazi.

**Investigation:** Laura Bahr, Francis RG Amrit, Paige Emily Silvia, Bella Wayhs, Guled Ali Osman, Mayur Nimbadas Devare, Hannah Henry, Danny Bui, Mirae Choe, Nikki Naim, Margaret Champion, Yuxuan Man, Arjumand Ghazi.

**Methodology:** Arjumand Ghazi.

**Project administration:** Arjumand Ghazi.

**Supervision:** Carissa Perez Olsen, Arjumand Ghazi.

**Writing – original draft:** Laura Bahr, Arjumand Ghazi.

**Writing – review & editing:** Laura Bahr, Guled Ali Osman, Mayur Nimbadas Devare, Carissa Perez Olsen, Arjumand Ghazi.

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
