## [Decision Letter · Decision Letter 0]

13 Aug 2025

PGENETICS-D-25-00799

LIPL-1 and LIPL-2 are TCER-1-regulated Lysosomal Lipases with Distinct Roles in Immunity and Fertility

PLOS Genetics

Dear Dr. Ghazi,

Thank you for submitting your manuscript to PLOS Genetics. After careful consideration, we feel that it has merit but does not fully meet PLOS Genetics's publication criteria as it currently stands. Therefore, we invite you to submit a revised version of the manuscript that addresses the points raised during the review process. 

Please submit your revised manuscript within 30 days Sep 12 2025 11:59PM. If you will need more time than this to complete your revisions, please reply to this message or contact the journal office at plosgenetics@plos.org. Please include the following items when submitting your revised manuscript:

We look forward to receiving your revised manuscript.

Kind regards,

Jeremy Nance

Academic Editor

PLOS Genetics

Pablo Wappner

Section Editor

PLOS Genetics

Aimée Dudley

Editor-in-Chief

PLOS Genetics

Anne Goriely

Editor-in-Chief

PLOS Genetics

**Journal Requirements:**

At this stage, the following Authors/Authors require contributions: Paige Emily Silvia. Please ensure that the full contributions of each author are acknowledged in the "Add/Edit/Remove Authors" section of our submission form.

The list of CRediT author contributions may be found here: https://journals.plos.org/plosgenetics/s/authorship#loc-author-contributions

https://journals.plos.org/plosgenetics/s/submission-guidelines#loc-parts-of-a-submission

4) We noticed that you used the phrase 'data not shown' in the manuscript. We do not allow these references, as the PLOS data access policy requires that all data be either published with the manuscript or made available in a publicly accessible database. Please amend the supplementary material to include the referenced data or remove the references.

5) Please upload all main figures as separate Figure files in .tif or .eps format. For more information about how to convert and format your figure files please see our guidelines: 

6) We notice that your supplementary Figures, and Tables are included in the manuscript file. Please remove them and upload them with the file type 'Supporting Information'. Please ensure that each Supporting Information file has a legend listed in the manuscript after the references list.

7) Please ensure that the funders and grant numbers match between the Financial Disclosure field and the Funding Information tab in your submission form. Note that the funders must be provided in the same order in both places as well.

**Reviewers' comments:**

Reviewer's Responses to Questions

**Comments to the Authors:**

Reviewer #1: The study by Bahr et al. identifies LIPL-1 and LIPL-2 as lysosomal lipases functioning downstream of TCER-1, a transcriptional and splicing regulator that promotes fertility while suppressing immunity in C. elegans. Through transcriptomic, lipidomic, and genetic analyses, the authors demonstrate that these lipases have distinct, context-dependent roles in immunity, reproduction, lipid metabolism, and lifespan: LIPL-1 enhances immune resistance, whereas LIPL-2 primarily supports fertility. Importantly, the human ortholog LAL rescues lipl-1–associated immune defects, suggesting an evolutionarily conserved role in host defense. This work advances our understanding of conserved lipases in regulating immunity and development.

I have several suggestions/questions to further improve the manuscript:

1. Are the translational reporters used in this study functional? Could the different promoter lengths used in various constructs influence the observed expression patterns?

2. The study reports lipl transgene expression in multiple tissues. In which tissues are lipl-1 and lipl-2 expression required and sufficient to produce the observed phenotypes?

3. Can supplementation with any lipid species identified in the lipidomics analysis rescue the infection or reproductive phenotypes in the mutant backgrounds?

4. Can the human homolog hLAL functionally replace lipl-2 as well? Does it also support reproductive function?

5. Adding a model figure summarizing the effects of each gene on immunity, reproduction, and lifespan would help synthesize the findings for readers.

6. There are inconsistencies in labeling double and triple mutants; the order of gene names should follow their chromosomal positions. tcer-1 is on chromosome II, while lipl-2 and lipl-1 are on chromosome V, with lipl-2 located earlier on the genetic map and therefore listed first. Promoter nomenclature should follow standard conventions, using gene name plus a lowercase p rather than a capital P.

7. Replace “normal” animals with “uninfected” animals for more precise language.

8. Remove references to “data not shown,” as PLOS does not permit this.

Overall, this is an interesting and well-executed study. Addressing the points above would improve clarity, reproducibility, and adherence to journal standards.

Reviewer #2: In this manuscript, Bahr et al. find that the expression of two lysosomal lipases, lipl-1 and lipl-2, changes in tcer-1 mutant animals, and show that these two genes have distinct activities and contribute to different tcer-1 mutant phenotypes. The authors identified lipl-1 and lipl-2 using RNAseq of tcer-1 mutants that were challenged with pathogenic Pseudomonas aeroginosa (PA). The abundance of the two lipl transcripts increase in wild-type animals challenged with PA. The induction of lipl genes is exaggerated in tcer-1 mutants, even though basal expression on non-pathogenic OP50 is reduced. The authors show that lipl-1 and lipl-2 mRNA are expressed in the intestine but that these proteins localize to coelomocytes, consistent with previous reports and studies of other lipl genes in C. elegans. Phenotypic analysis shows that lipl-1 is required for the increased survival of tcer-1 mutants when challenged with PA but does not change the survival of wild-type animals on PA. In contrast, loss of lipl-2 itself confers resistance to PA in wild-type animals similar to tcer-1 and there is no additive effect in the tcer-1; lipl-2 double mutant. The effects of lipl genes is specific to PA, as the lifespan of single and double mutants on OP50 was indistinguishable from wild-type. The authors also investigated the role of these lipl genes in fertility, as tcer-1 mutants are known to have reduced fertility. The authors show that loss of either lipl-1 or lipl-2 further reduces the number of progeny produced by tcer-1 mutants, but have no effect in the wild-type background. Closer examination showed that loss of either lipl gene increased embryonic lethality and sterility in tcer-1 mutants, and that loss of lipl-2 (but not lipl-1) increases the fraction of embryos with an intact permeability barrier. The authors next examine how the loss of the lipl genes changes the lipid profile of tcer-1 animals. The lipl genes are orthologues of the human lysosomal acid lipase (hLAL), and the authors show that hLAL can rescue the immunity defect of lipl-1 animals, suggesting some functions are conserved.

Previous work from this group has demonstrated that TCER-1, an orthologue of the human transcription elongation and splicing factor TRCERG1, coordinates immunity, longevity, and fertility. This work shows that the effects of TCER-1 on immunity and fertility are mediated, in part, by regulation of LIPL-1 and LIPL-2. Moreover, the authors demonstrate that these two lipases have distinct physiological effects. This is an important finding that has implications in our understanding of mechanisms that underlie innate immunity, fat metabolism, reproductive fitness, and how these processes are coordinated. The data are rigorous and clearly presented, and this work will be of interest across fields.

I think that this work would be even stronger if the authors could more clearly show that the changes they observe were functionally relevant. A couple of examples:

* It is not clear if observed changes in embryo permeability are related to the fertility defect; whereas increased permeability is only observed in the tcer-1;lipl-2 double mutant, embryo viability, sterility, or brood size are all similarly reduced by lipl-1 and lipl-2. Moreover, the fraction of embryos permeable to dye (1.5% according to Fig S4) is much lower than the fraction of embryos that don’t hatch shown in Fig 5C. While it is certainly interesting that lipl-2 has this effect on permeability, It seems like there could be other tcer-1-associated changes that may be more germane to the effect on fertility observed. How do the authors explain the lack of correlation between permeability and the fertility defect they observe? Is it possible that the reduced brood size (Fig 5A) is reflecting a defect in sperm production or function? Does PA14 exposure exacerbate the permeability effect?

* The authors show that there are some changes in the lipid profile of tcer-1 are related to lipl-1 and/or lipl-2, but there is no data to indicate which, if any, of these changes are functionally relevant. Neither lipl-1 nor lipl-2 impact the DAG or TAG of tcer-1 mutants (Fig 6A,B), suggesting that these are not relevant to the phenotypes of the double mutants. In contrast, there are some tcer-1-associated changes in lipids that do seem to be differently impacted by lipl-1 and lipl-2 (such as ceramides Cer17:1; O2/22:0 and Cer17:1; O2/24:0). It would be interesting to know if any of the tcer-1 immunity or fertility phenotypes are sensitive to orthogonal perturbations in these lipids. It would also be interesting to know how the authors are thinking about the pattern of lipid changes - why do some ceramides change, but not others? Why are some unsaturated lipids more affected than others?

I would also like to raise some minor points that I hope the authors could address that I believe would improve the manuscript:

* It’s not entirely clear to me how the authors identified the lipl genes from the RNAseq data presented. The way the manuscript is presented, it seemed that the authors were most interested in tcer-1-dependent, PA14 responsive transcriptional changes but it is a bit confusing which set of differentially expressed genes (DEG) they include in this set. The text suggests that the genes of interest would be included in the WT/PA14 vs tcer-1/PA14. However, some of these genes (including lipl-1 and lipl-2) are also included in the tcer-1/OP50 vs tcer-1/PA14 DEG, which the authors indicate are “tcer-1 independent” transcriptional responses. It could also be argued that the DEG in the wild-type PA14 response (WT/OP50 vs WT/PA14) that do not change in tcer-1 (OP50 vs PA14) are also “tcer-1-independent” responsive.

* Do any of the other 6 lipl genes change in response to PA14?

* I don’t understand why the authors didn’t use the DEG in PA14 that were tcer-1 dependent (WT/PA14 vs tcer-1/PA14) in the wormcat analysis shown in Fig 1G. The way these data are presented, there does not seem to be much difference in the WT and tcer-1 responses. It is really not clear how this result helped to identify the lipl genes.

* It seems that in Fig 1I the authors statistically compare lipl-2 expression between WT/PA14 and tcer-1/OP50. It is not clear why this comparison would be considered instead of analyzing these data the same as for lipl-1 in Fig 1H.

* The authors compare these RNAseq data to previous published transcriptomic analysis of PA14 responses. While there are significant overlaps in the genes that change in each data set, as expected, there are a lot of changes that do not overlap. Do the authors have an idea for why these data sets are so different?

* The authors suggest that the LIPL reporters are expressed in the head of a small fraction of worms. I’m not convinced that this reflects the normal expression pattern of these genes, and I think including this observation in the main figures is somewhat misleading. Unless the authors want to do more experiments with relevant controls, I would recommend that they do not include pictures of these rare animals in the figures. Similarly, the author claim that this low and uncommon expression is in neurons but it seems that it might also be hypodermis or pharynx. This should be corrected unless the authors can show colocalization with a neuronal marker.

* The authors show images of a single worm to support their claims in Fig 2, SFig 2, Fig 3, and SFig 3. Especially when there is not a quantitation of a population, showing a single worm is not sufficient of the reader to evaluate if these data support the stated conclusions.

* It is not clear how many independent replicates were done for lifespan experiments, or if the Figure shows a single experiment or if it is an aggregation of all replicates. This should be made clear in the figure legend, and summary statistics for all replicates should be included in the Supplementary Materials (mean survival, max survival, # animals, # censored, etc).

* In Fig 5, the authors claim to measure the “hatch rate”, but this is not a rate measurement, it is just the fraction of the eggs that hatch. It is also not clear how the authors defined “sterility” for the data shown in Fig 5B.

* It is not clear how many replicates are included in the lipidomics experiments and this makes it difficult to interpret the data. Many of the changes seem pretty small, and it would be useful to know how the abundance varies across samples to have a better understanding of the robustness of the changes that are noted.

* Is the hLAL protein also localized to the coelomocytes? Other work suggests that localization to coelomocytes is important for LPL function (Buis et al 2019) - is this true for tcer-1 associated effects of lipl-1 and/or lipl-2?

* On page 20 it would be good to include the relevant reference where the plipl-2::LIPL-2::RFP transgenic strain was first characterized.

* There are some unformatted citations on pg 24, 28, and 29.

* In the materials and methods the authors should replace “u” with the greek mu where appropriate

* For pathogen exposure experiments, how many independent replicates were performed? Is the data shown in the figures aggregated data or a representative experiment? The authors should include this information in the legends, and also provide summary statistics for independent replicates in the supplementary material. Also, the authors mention S. aureus in the methods section but I don’t think any of these experiments is included in this manuscript.

* The authors used FuDR to prevent egg hatching on PA14, but don’t show controls to make sure that FuDR doesn’t affect the results of these assays. It would be nice to see this control.

* For experiments comparing transgene expression, what calibrations were performed to ensure that comparisons between samples are valid? Also, were the samples blinded in any way before the assay?

* In the materials and methods the authors suggest that they separated young adults from other stages for lipidomics experiments with strains that were not easily synchronized. Was this done manually? It is not clear what exactly was done to achieve this.

Reviewer #3: Using the C. elegans model, Bahr et al. investigate the role of two lipases, LIPL-1 and -2 in the regulation of life span, immune responses, and reproduction, both in wild type worms in the tcer-1 mutant background. Deploying lipidomic, trascriptomic, and genetic (mutant/rescue and reporter) approaches they find that these two lipases are regulated substantially in tcer-1 mutants and after infection with the Pseudomonas aerigunosa strain PA14. Despite the similar regulation of both genes, their loss of function causes disparate effects, with lipl-1 required for the enhanced resistance of tcer-1 to pathogen, and tcer-2 for the life span phenotype. Both genes showed defects, and synthetic effects on various parameters of reproduction. Excitingly, lipidomics revealed some distinct changes between lipl-1 and -2, for example the blocked induction of 3 ceramide species by lipl-1 mutation in the tcer-1 context. Finally, the authors showed that the human LIPL ortholog, hLAL, could rescue the defective pathogen resistance caused by lipl-1 mutation.

This study provides important new insight into both the phenotypes and underlying molecular lipid defects of two lipases and should be of interest to researchers studying gene regulation, metabolism, immune responses, and reproduction. The experiments are overall well done and very comprehensive.

Comments:

- It would be exciting to be able to show whether the deregulated ceramide species can be supplemented to rescue the defects in immune defence; I appreciate that this may be a challenging experiment however.

- I was surprised that the authors didn’t also attempt to test whether hLAL also rescues lipl-2 mutant phenotypes

- Results P13: conclusion "lipl-2 single mutants also showed a similar change in TAG:DAG distribution which was further aggravated in tcer-1;lipl-2 double mutants.” Differences between the double and the singles appear minor and not stats are shown - did the authors mean to write distribution was NOT further aggravated? I would reason that that is what’s supported by the data and statistics

Typos

- Abstract: "lipl-l”

- Intro p4: “Arabidopsis” add “thaliana”

- Results p9: "LIPL-1::RFP expression showed similar spatial distribution as the mRNA” - GFP reporters measure promoter activity, not mRNA. Rephrase.

- Results p12: "the lipl-1 or lipl-1 mutations”

- Results p12 “even reproductively older Day 4” sounds odd - aren’t they simply “older”?

- Results p15 "LPC 18: 2” delete space

- P18 discussion: “lOur"

**Have all data underlying the figures and results presented in the manuscript been provided?**

Reviewer #1: None

Reviewer #2: Yes

Reviewer #3: **No: ** RNA-seq data should be deposited in GEO or similar. Lipidomics data should be deposited in a similarly accessible databank

PLOS authors have the option to publish the peer review history of their article (what does this mean? ). If published, this will include your full peer review and any attached files.

**Do you want your identity to be public for this peer review?** For information about this choice, including consent withdrawal, please see our Privacy Policy .

Reviewer #1: No

Reviewer #2: No

Reviewer #3: No

**Figure resubmission:**
---

## [Decision Letter · Decision Letter 1]

18 Nov 2025

Dear Dr Ghazi,

We are pleased to inform you that your manuscript entitled "LIPL-1 and LIPL-2 are TCER-1-regulated Lysosomal Lipases with Distinct Roles in Immunity and Fertility" has been editorially accepted for publication in PLOS Genetics. Congratulations!

Yours sincerely,

Jeremy Nance

Academic Editor

PLOS Genetics

Pablo Wappner

Section Editor

PLOS Genetics

Aimée Dudley

Editor-in-Chief

PLOS Genetics

Anne Goriely

Editor-in-Chief

PLOS Genetics

BlueSky: @plos.bsky.social

Comments from the reviewers (if applicable):

Reviewer's Responses to Questions

**Comments to the Authors:**

Reviewer #1: I would like to thank authors for addressing all my comments/questions.

Reviewer #2: The authors have addressed every point that I raised, and the issues raised by the other reviewers, and I think the manuscript is improved. I recommend it be accepted for publication.

Reviewer #3: The reviewers have addressed all my comments to satisfaction. Congrats on a very nice paper!

**Have all data underlying the figures and results presented in the manuscript been provided?**

Reviewer #1: None

Reviewer #2: Yes

Reviewer #3: **No: ** Not sure if large scale RNA-seq and lipidomics datasets have been deposited in relevant databases

PLOS authors have the option to publish the peer review history of their article (what does this mean? ). If published, this will include your full peer review and any attached files.

**Do you want your identity to be public for this peer review?** For information about this choice, including consent withdrawal, please see our Privacy Policy .

Reviewer #1: No

Reviewer #2: No

Reviewer #3: No

**Data Deposition**

http://datadryad.org/submit?journalID=pgenetics&manu=PGENETICS-D-25-00799R1

**Press Queries**

---

## [Editor Report · Acceptance letter]

PGENETICS-D-25-00799R1

LIPL-1 and LIPL-2 are TCER-1-regulated Lysosomal Lipases with Distinct Roles in Immunity and Fertility

Dear Dr Ghazi,

We are pleased to inform you that your manuscript entitled "LIPL-1 and LIPL-2 are TCER-1-regulated Lysosomal Lipases with Distinct Roles in Immunity and Fertility" has been formally accepted for publication in PLOS Genetics! Your manuscript is now with our production department and you will be notified of the publication date in due course.

With kind regards,

Narmatha Raju, M.Sc

PLOS Genetics

On behalf of:
